
**A Procedure to Select Earthquake Time Histories for Deterministic Seismic Hazard Analysis:**
**Case Studies of Major Cities in Taiwan**
Duruo Huang[1] and Wenqi Du[2]
[1]Department of Civil & Environmental Engineering, the Hong Kong University of Science and Technology,
Kowloon, Hong Kong.
[2]Institute of Catastrophe Risk Management, Nanyang Technological University, Singapore.
*Correspondence to*: Duruo Huang (huangdr@ust.hk)
**Abstract:** In performance-based seismic design, ground-motion time histories are needed for analyzing
dynamic responses of nonlinear structural systems. However, the number of strong-motion data at
design level is often limited. In order to analyze seismic performance of structures, ground-motion time
histories need to be either selected from recorded strong-motion database, or numerically simulated
using stochastic approaches. In this paper. a detailed procedure to select proper acceleration time
histories from the Next Generation Attenuation (NGA) database for several cities in Taiwan is presented.
Target response spectra are initially determined based on a local ground motion prediction equation
under representative deterministic seismic hazard analyses. Then several suites of ground motions are
selected for these cities using the Design Ground Motion Library (DGML), a recently proposed
interactive ground-motion selection tool. The selected time histories are representatives of the regional
seismic hazard, and should be beneficial to earthquake studies when comprehensive seismic hazard
assessments and site investigations are yet available. Note that this method is also applicable to site-
specific motion selections with the target spectra near the ground surface considering the site effect.
**Keywords:** Ground motion selection, Seismic hazard analysis, NGA database, DGML tool



## 1 Introduction

In performance-based earthquake engineering, ground-motion time histories are usually needed for
analyzing the distribution of dynamic responses of nonlinear systems, such as site response or structural
analysis. In such an analysis, it is one of the key aspects to use appropriate acceleration time histories,
which should realistically reflect regional seismology and site conditions.
Understandably, the selected time histories should reasonably respond to seismic hazards at a given
site. For example, a recent technical guideline implemented by the U.S. Nuclear Regulatory
Commission (USNRC, 2007) prescribed the probabilistic seismic hazard analysis (PSHA) as the
underlying approach to generate time histories for future earthquake-resistant designs. Many studies
have highlighted the importance of matching a target response spectrum in the ground-motion selection
and modification process (e.g., Bommer and Acevedo, 2004). The target spectrum can be obtained by
deterministic seismic hazard analysis (DSHA), probabilistic seismic hazard analysis (PSHA) or seismic
design codes. A classic example is SIMQKE, which generates synthetic time histories to match a target
response spectrum with an iterative process using Gaussian random process and a time-varying
modulating function (Gasparini and Vanmarcke, 1976).
Recently, some scholars studied that a well-selected ground motion suite should match not only the
target mean, but also the variation of the target spectrum (Jayaram et al., 2011; Wang, 2011). In other
words, a suite of ground motions should be selected in performance-based earthquake engineering; the
resulting ground motion suite should properly capture the statistical distribution of ground motions
under the given earthquake scenario, which is commonly specified by means, standard deviations, and
inherent correlations (e.g., Baker and Jayaram, 2008; Wang and Du, 2012) of a target spectrum. There
are several ground motion selection algorithms available in the literature (Baker, 2010; Jayaram et al.,
2011; Wang, 2011). One of the recently proposed interactive tools is the Design Ground Motion Library
(DGML), which allows for selecting a suite of modified ground motions (multiple by scale factors) on
the basis of response spectral shape, as well as the characteristics of the recordings such as magnitude,
distances, faulting types and site conditions (Wang et al., 2015).
This paper aims at presenting a detailed procedure in selecting ground-motion time histories for
major cities of Taiwan using the DGML interactive tool. With deterministic seismic hazard analysis for





these cities, several suites of time histories are selected from the Pacific Earthquake Engineering Research Center's Next Generation Attenuation (NGA) strong-motion database (Chiou et al., 2008). Those selected motion suites are appropriate for general seismic designs, e.g., dynamic analysis of structures in these cities.

## 2   Deterministic Seismic Hazard Analyses (DSHA) for Major Cities in Taiwan

### 2.1 Overview of DSHA

Seismic hazard analysis is an approach to describe the potential shaking intensity for future earthquakes, which can be estimated by deterministic or probabilistic approaches. The deterministic approach estimates the intensity measure amplitude (e.g., peak ground acceleration PGA as 0.2 g) under an assigned earthquake scenario, while the probabilistic approach estimates the annual rate of exceeding specific level of earthquake shaking at a site (e.g., PGA=0.2 g corresponding to 10% probability of exceedance in 50 years).

Compared to the complicated probabilistic approach, DSHA is a logically simple and transparent method. The purpose of DSHA is to use the maximum magnitude and shortest source-to-site distance to evaluate the ground motion intensities under such a worse-case scenario. The basic steps are listed as follows: (1) Identify all possible fault sources of earthquakes around a given site; (2) Define the maximum magnitude and closest distance for each fault; (3) Compute the ground motion intensities based on attenuation relationships; (4) Take the maximum intensity amplitudes as the final DSHA estimate. Figure 1 shows a schematic diagram illustrating the framework and the algorithm for DSHA. Seismic source models, the maximum earthquake of each source, and ground motion prediction equations (GMPEs) are key inputs for DSHA. The detailed source models and GMPEs used in this study would be introduced in this following subsection.

### 2.2 Seismic source model and ground-motion model

Figures 2 and 3 show the up-to-date seismic source models for Taiwan (Cheng et al., 2007), which have also been used in a few seismic hazard studies by several authors (Cheng et al., 2007; Wang and Huang, 2014). It includes 20 area sources, in addition to 49 line sources associated with each active fault on





this island. Table 1 summarizes the best-estimated maximum magnitude for each source from the
literature (Cheng et al., 2007). With those best estimates, the response spectra for major cities in Taiwan
are also presented in this section with DSHA calculations.

Ground motion prediction equations (GMPEs) are commonly used to predict ground motion

intensities (e.g., PGA) as a function of earthquake magnitude, source-to-site distance, site parameters,
etc. A few regional GMPEs models have been developed based on local strong-motion data in Taiwan
(Cheng et al., 2007; Lin et al., 2011). Specifically, the recent GMPE developed by Lin et al. (2011) is
capable of predicting PGA and response spectra for periods ranging from 0.01 s to 5 s, and therefore it
is adopted in this study, to develop the target response spectra for selecting earthquake time histories.

The function form of the adopted model (Lin et al. 2011) is expressed as follows:

$$\ln Y = c_1 + c_2 M_w + c_3 \ln(R + c_4 e^{c_5 M_w}) \qquad \sigma_{\ln Y} = \sigma * \qquad (1)$$

where $Y$ denotes PGA or spectral accelerations in unit of $g$; $M_w$ refers to moment magnitude; $R$ is the
rupture distance (closest distance from the rupture surface to site) in $km$; $c_1$ to $c_5$ are regressed
coefficients. The model's coefficients are summarized in Table 2, and $\sigma_{\ln Y}$ denotes the model's standard
deviation. It is noted that this model was developed using around 5,000 earthquake records, 98% of
which are taken from Taiwan. Therefore, the attenuation model should provide more realistic ground
motion estimates in Taiwan (Lin et al., 2011), making it appropriate to construct the target response
spectra.

It is also worth noting that we only employ the local ground motion model in this study. It is

understood that logic-tree analyses can be used to quantity the so-called epistemic uncertainty in PSHA.
But as studied by some scholars (e.g., Krinitzsky, 2003), the weights in logic-tree analyses cannot be
scientifically verified. Therefore, this study used one local model available as the best estimate. When
new local models are developed, the update of seismic hazards or sensitivity analyses will be worth
conducting in future.


**2.3 DSHA-based response spectra for major cities in Taiwan**



The aforementioned DSHA procedures can be performed for major cities in Taiwan, with the adopted
seismic source models (Figures 2 and 3) and attenuation relationship introduced in previous subsections.
Six major cities are chosen for such calculations, and coordinates of the study cities (i.e., the city's
geographical centers) are summarized in Table 3. For each site or city, the worse-case scenario was
firstly identified, and then the corresponding response spectrum was determined by using the adopted
local GMPE.
Figure 4 shows the resulting response spectra from DSHA calculations for the six considered cities
in Taiwan. Table 3 also summarizes the controlling seismic source for each site. For example, the
DSHA seismic hazard at the center of Taipei is governed by Area Source C. In other words, the Area
Source C, rather than the other line sources or active faults, contributes to the deterministic seismic
hazard for the center of Taipei. The same situation is occurring to other cities with an area source being
the controlling source. This is expected, since the DSHA seismic hazard from an area source could be
commonly higher than a line source due to the relatively closer source-to-site distance.
It should be noted that the adopted local GMPE has been thoroughly compared with the globally
NGA GMPEs (Abrahamson and Silva, 2008; Boore and Atkinson, 2008; Campbell and Bozorgnia,
2008; Chiou and Youngs, 2008). In general, the PGA amplitudes predicted by the adopted model is
generally comparable to those of the NGA models, except that for scenarios with distances greater than
20 km the estimated PGAs of the local model attenuate faster. The steeper slope of the local attenuation
curves could be due to the fact that the local crust is relatively weak, given that Taiwan is a very young
orogeny (Lin et al., 2011). This implies that a design or target spectrum derived from local GMPEs is
particularly necessary for selecting suitable ground-motion time histories for local engineering practice.
**3  Selection of Ground-Motion Time Histories**
**3.1 The NGA database and Design Ground Motion Library (DGML)**
The source for ground-motion selection in this study is the PEER-NGA strong motion database, which
contains 3,551 three-component recordings from 173 earthquakes (Chiou et al., 2008). Various subsets
of the database have been used to develop GMPE models for various ground motion intensities in
earthquake engineering (e.g., Du and Wang, 2013; Foulser-Piggott and Stafford, 2012). Figure 5 shows




the moment magnitude-rupture distance distribution of the ground motions in the NGA database. The
aforementioned interactive tool, DGML, is used to search ground-motion time histories in the NGA
database on the basis of similarity of a record's response spectrum to the target response spectrum over
a use-defined range of period (Wang et al., 2015). The DGML has the broad capability of searching for
ground-motion time histories in the library database on the basis of response spectral shape,
characteristics of the recordings in terms of earthquake magnitude and type of faulting distance, site
characteristics, duration, and presence of velocity pulses in near-fault time histories.
To select appropriate ground motions by DGML, it is requested to specify the seismological
parameter bounds (e.g., range of considered $M_w$ and distance $R$) as inputs, which can implicitly
constrain the ground motion characteristics in addition to the explicit target spectrum. Given the fact
that the target spectra from DSHA are a result of the maximum earthquake and the closest source-to-site
distance, a relatively large magnitude bound ($5.5<M_w<8$) and a narrow distance range ($0$ km$<R_{rup}<30$
km) have been employed as the searching criteria, as shown in Fig. 6. Since all the six cities are located
at soil sites, a $V_{s30}$ (time-averaged shear-wave velocity down to 30 m) bound in the range of 0-450 m/s
is also applied. Other causal parameters, such as the category of fault types or the range of duration
parameters, are not particularly specified.
Scaling factor is another key input for selecting ground motions. In engineering practice, recorded
ground motions usually need to be up-scaled to the level of the target or design spectrum. It has been
studied that time histories scaled by an appropriate factor could lead to an acceptable response results
(Watson-Lamprey and Abrahamson, 2006). Yet, if an excessive range of scale factors is applied, the
selected ground motion suite might result in drastically biased distribution of the other ground motion
characteristics (e.g., duration parameters) that cannot be represented by the target response spectrum.
Therefore, a relative narrow range of scale factors (0.4-2.5) is applied in this selection procedure.
Figure 6 shows the interface of DGML while searching for properly matched time histories with
target spectrum and magnitude and distance thresholds. The ranking of earthquake motions is tabulated
after spectral matching process. The motions of interest can be downloaded from the list, as well as their
descriptions such as fault types, earthquake magnitudes, rupture distances, durations, scaling factors,
and $V_{s30}$ values ($V_{s30}$ is commonly employed site condition indicator). Note that DGML is also capable



of performing weight-matching when a specific range of the motion's frequencies is of more interest in
follow-up applications.
**3.2 Time history recommendations for major cities of Taiwan**
With the target spectra from DSHA calculations, the selection procedures in DGML are performed to
select a suite of time histories from the NGA database for each city. Figure 7 shows the selected
response spectra for the six study cities. The median and median ± one standard deviation of the
selected SA ordinates are also compared to the target spectrum in each plot. It can be seen that the
selected ground motion suites can properly match the target spectra over a wide period range. Table 4
summarizes the time histories selected from the database. Figures 8-14 show the selected time histories
for the six cities in Taiwan with seismic hazards calculated with DSHA calculations. Note that two sets
of selections were given for Taipei, with and without the consideration of basin effect. It should also be
noted that for each site the best-matching motions were selected regardless of local earthquakes or not,
in addition to one or two best-matching local motion (i.e., the Chi-Chi earthquake). The multiple time
histories in each suite are considered as a measure to account for the variability or natural randomness
of ground motion characteristics under a considered scenario, which, for example, is considered as
mandatory for probabilistic site response analyses prescribed in a technical reference (USNRC, 2007).
**4  Discussions**
**4.1 DSHA versus PSHA**
PSHA and DSHA are the two representative approaches in assessing earthquake hazards. Over the past
decades, numerous seismic hazard studies have been conducted with the two methods (e.g., Joshi et al.,
2007; Kolathayar and Sitharam, 2012; Moratto et al., 2007; Sitharam and Vipin, 2011; Stirling et al.,
2011). The two methods have also been prescribed in various technical references. As mentioned
previously, a technical reference (USNRC, 2007) prescribes PSHA as the underlying approach, in
contrast to another guideline implemented by Department of California Transportation prescribing
DSHA for bridge designs under earthquake loadings (Mualchin, 2011).





It is worth noting that extensive discussions over the pros and cons of the two methods have been
reported in the literature (e.g., Bommer, 2003; Castanos and Lomnitz, 2002; Krinitzsky, 2003; Klugel,
2008). In general, DSHA is a simple approach that earthquake scenarios are considered logically
understandably, but the uncertainties in DSHA may not be well quantified. On the other hand, PSHA is
capable of quantifying the uncertainties associated with earthquake scenarios via a probabilistic
approach; however, some scholars (e.g., Krinitzsky, 2003) pointed out the shortcomings in PSHA, such
as the uniform assumption in the occurrences of earthquakes. It is not this paper's purpose to argue
which seismic hazard method is superior. But with all that in mind, it should come to a logical
understanding that both the deterministic and probabilistic analyses are needed and useful in
engineering applications. The use of the DSHA approach in this paper is mainly due to its analytical
simplicity and transparency. Since it has been reported that DSHA rather than PSHA is more
appropriate for design of critical structures (Bommer et al., 2000), the selected ground motion suites,
with a representative seismic hazard analysis and a reputable earthquake database, are then
recommended for such applications.
**4.2 Site-specific time histories**
This paper presents an option to select earthquake time histories from the reputable NGA database. But
strictly speaking, those time history recommendations are not site-specific, because the site condition is
not carefully taken into account with a comprehensive site investigations and site response analyses. In
other words, the site-specific motions are those from seismic hazard analyses, to site response studies
(e.g., Du and Pan, 2016).
As a result, this study refers to those time-history recommendations as "tentative site-specific,"
because the site effect is not comprehensively characterized with a more detailed site response analysis,
but with a soil-site ground motion prediction model. Therefore, the selected ground motion time-
histories could be recommended for general earthquake analytical cases, where specific site
investigations are not performed. Since the recommended time-histories can reasonably reflect the local
seismic hazards at these cities, they should be used as basic results and then be serviceable for common
engineering practice.





### 4.3 Basin effect

Basin effect is another important issue to estimate the seismic hazards for sites within Taipei. From analyzing the recorded time histories around Taipei (Sokolov et al., 2009; 2010), some suggestions were made to up-scale low-frequency spectral accelerations to incorporate the basin effect in Taipei. Following this suggestion, Figure 15 shows the response spectra with/without considering basin effects for Taipei by DSHA calculations. Likewise, the time histories matching the up-scaled spectra (with basin effects) as the target are selected from the database, as summarized in Table 4.

### 4.4 Why Chi-Chi earthquake's motions are not selected?

It somewhat comes to as a surprise that the motions of the Chi-Chi earthquake were "out-performed" by non-local motions in matching the response spectra with local ground motion models. This is might be due to two reasons. First, the adopted local GMPE was developed with 42 earthquakes, 85% of which are not associated with the Chi-Chi earthquake, its foreshocks and aftershocks (Lin et al., 2011). Therefore, the influence of the Chi-Chi earthquake (or others) should not play a dominating role on the model performance, given such a pool of data employed. Except the Chi-Chi earthquake, most events used for developing the local GMPE are not included in the NGA database.

The second reason is that the employed searching process does not specify more weights or preferences to local earthquakes. As discussed previously, the search criterion are only associated with the spectral shape, as well as seismological parameters such as magnitude, distance, site condition, *etc*. With this in mind, as long as the size of the database is sufficient, it is not surprising that a non-local ground motion can be found better matching the target spectra. This could also be the reason that the NGA database features the functionality to perform limited searching among selected earthquakes, when local earthquakes are judged to be more suitable for an application.

### 5 Conclusions

The paper presented the procedures to select earthquake time histories with target response spectra from deterministic seismic hazard analysis (DSHA), using the recently proposed DGML selection tool. The worst-case earthquake scenarios were first defined for six major cities in Taiwan, and the response





target spectra were computed by employing a regional attenuation model under these defined scenarios.
Finally, a suite of time histories are selected for each city by matching the calculated target spectra. The
selected suites of time histories can properly represent the regional seismic hazards, which are then
recommended and used for seismic analyses in these cities. The similar ground motion selection
approaches can also be applicable to selecting appropriate time histories at bedrock layers, as input
motions for a more comprehensive site investigations and site response analysis.
Given the limited understandings of the earthquake process and the randomness in nature, some
scholars have pointed out the importance of analytical simplicity to earthquake studies. Among several
approaches to define the target spectra, the ones from DSHA calculations are logically transparent and
simple, and therefore they are adopted in this study for selecting hazard-consist time histories.
**Acknowledgments**
The authors acknowledge financial supports provided by the Ministry of Home Affairs and the
Monetary Authority of Singapore for this work.

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



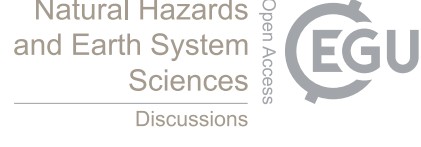

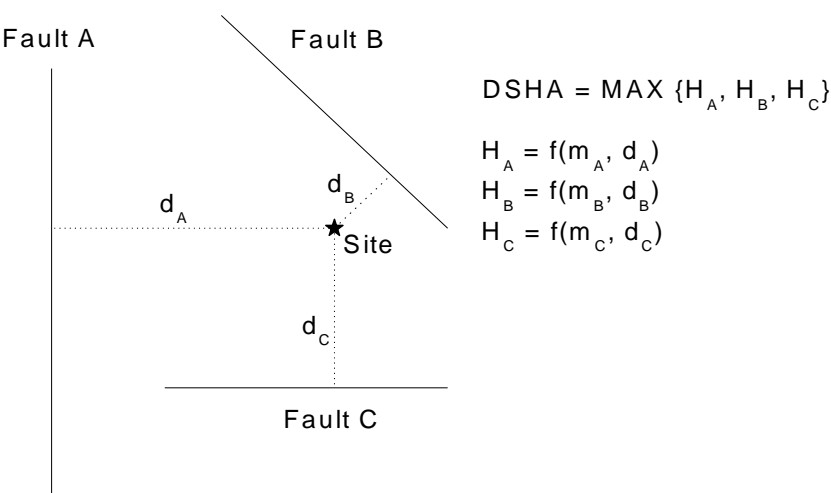

**Figure 1.** Schematic diagram illustrating the analytical framework of DSHA, where *H* denotes the seismic hazard induced by each source, *m* and *d* are the maximum earthquake magnitude and shortest source-to-site distance, and *f* is the function of a ground motion model





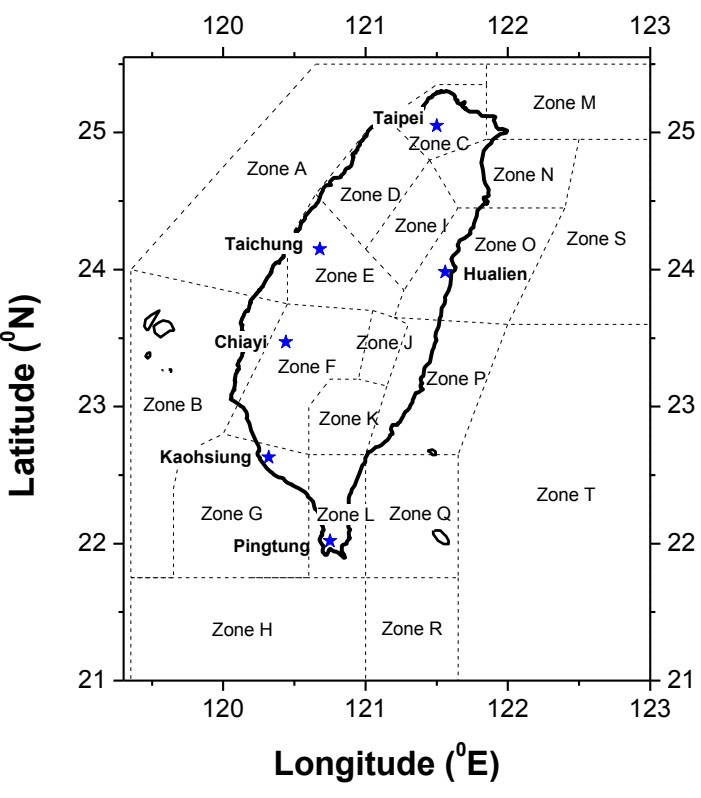

**Figure 2.** The area seismic source model for Taiwan (after Cheng et al., 2007)





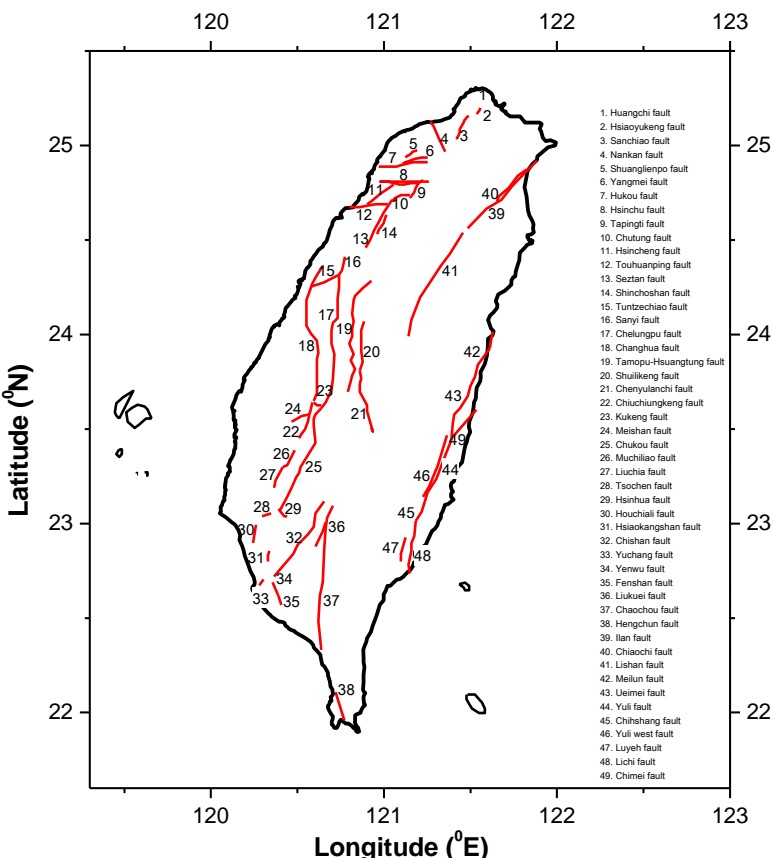

**Figure 3.** The line source model or the active faults in Taiwan (after Cheng et al., 2007)



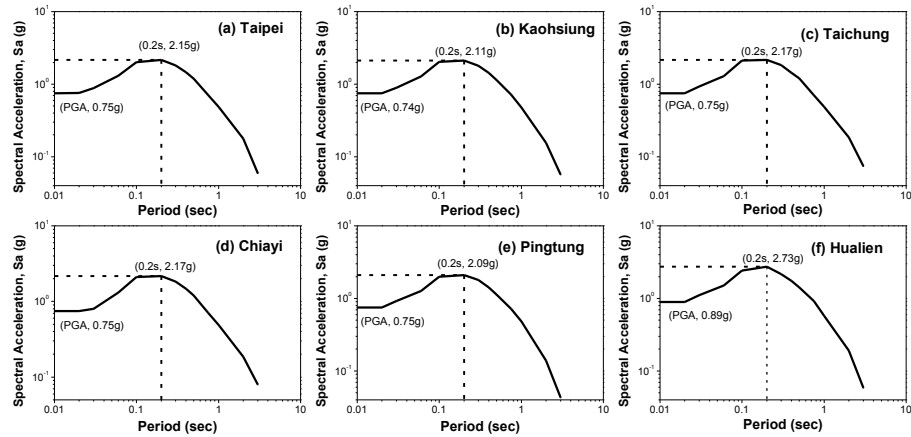

**Figure 4.** The response spectra for major cities in Taiwan with DSHA calculations





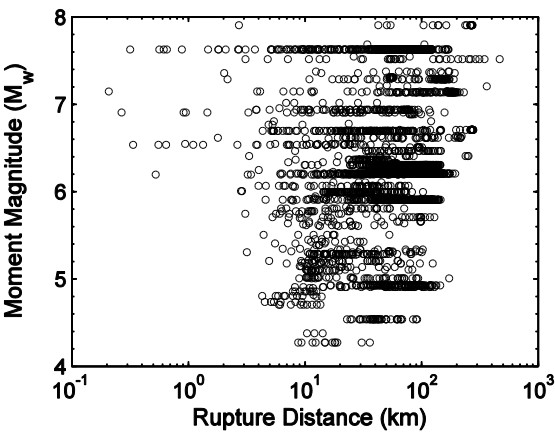

**Figure 5.** Moment magnitude and rupture distance distribution for PEER NGA records used in this study





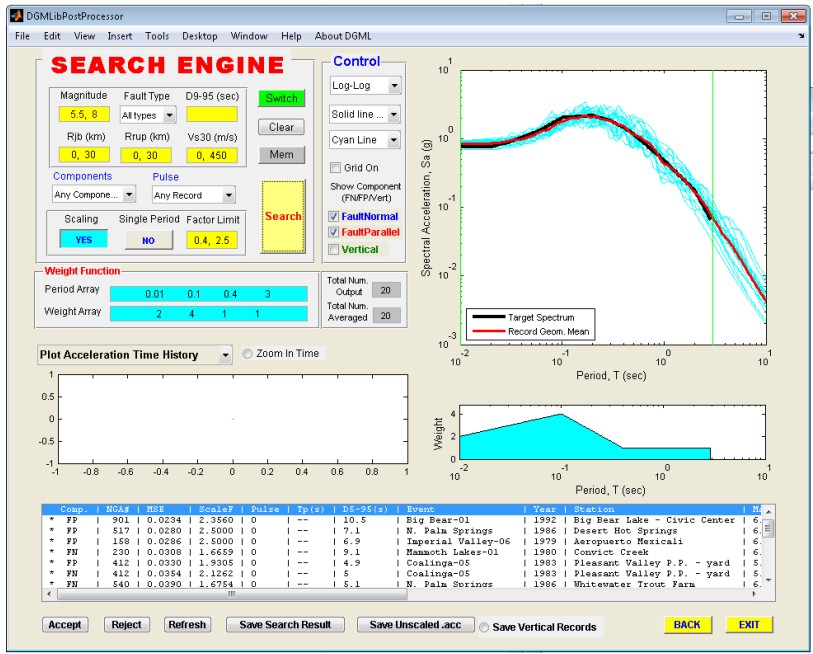

**Figure 6.** The screenshot of the database's interface; with searching criteria as shown in the left, the properly matching motions are tabulated (not shown), and their response spectra are plotted in a graph along with the target spectra, shown in the right





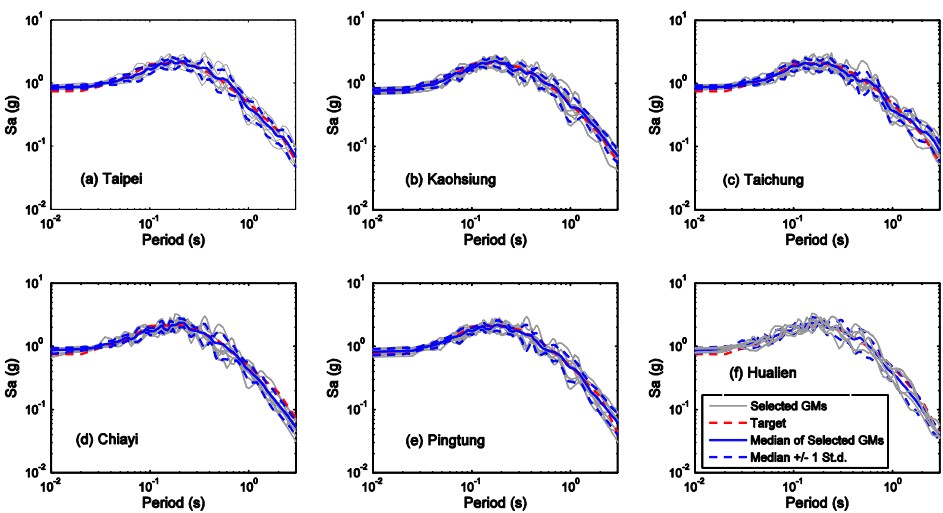

**Figure 7.** The target spectrum, individual and average response spectrum of selected records for six major cities in Taiwan




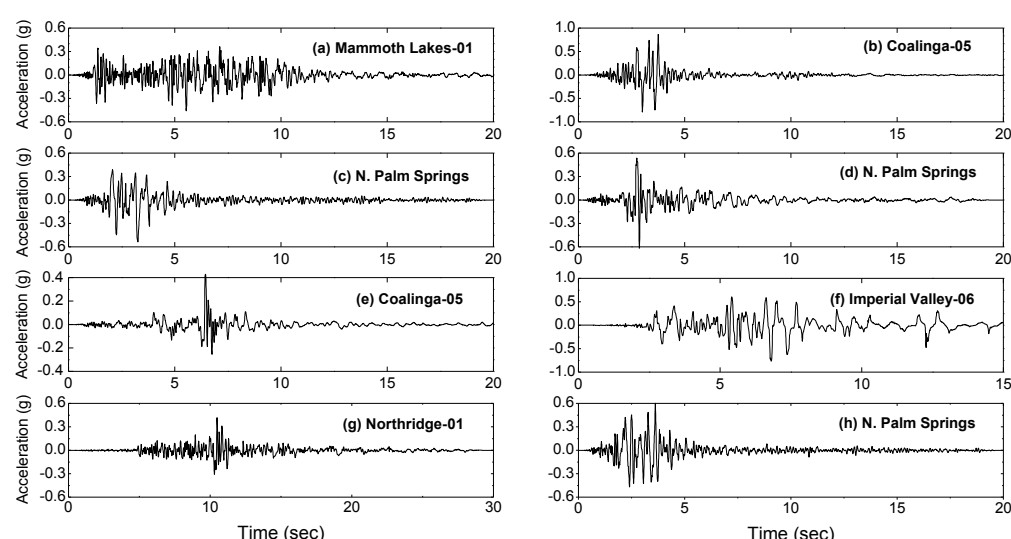

**Figure 8.** Eight time history recommendations for Taipei with DSHA calculations and the NGA strong-motion database





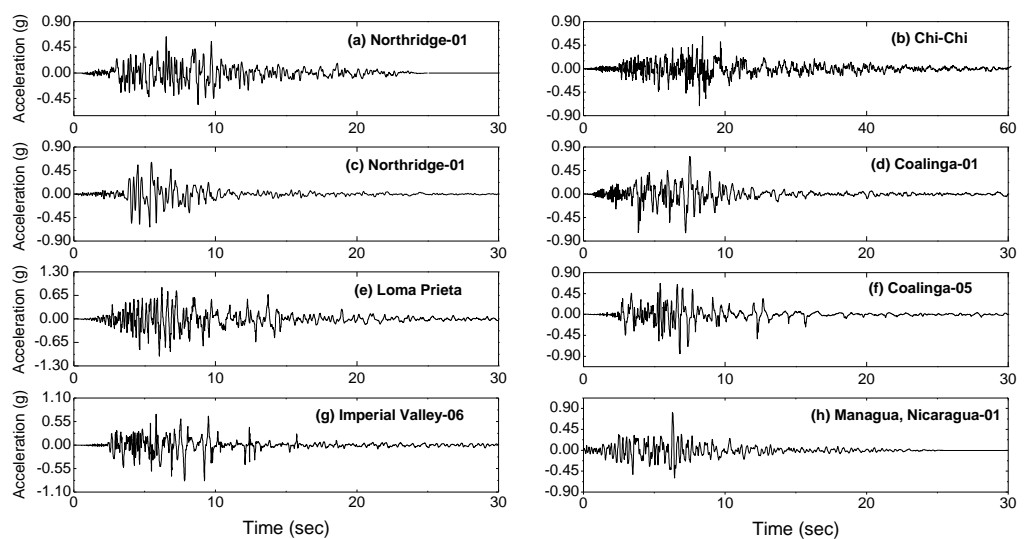

**Figure 9.** Another set of time history recommendations for Taipei with the basin effect taken into account





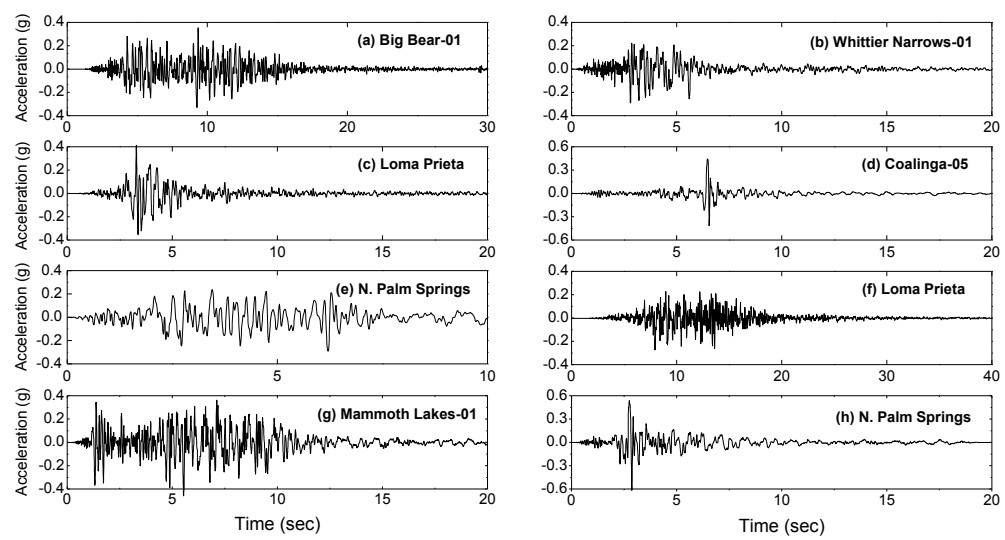

**Figure 10.** Eight time history recommendations for Kaohsiung with DSHA
calculations and the NGA strong-motion database





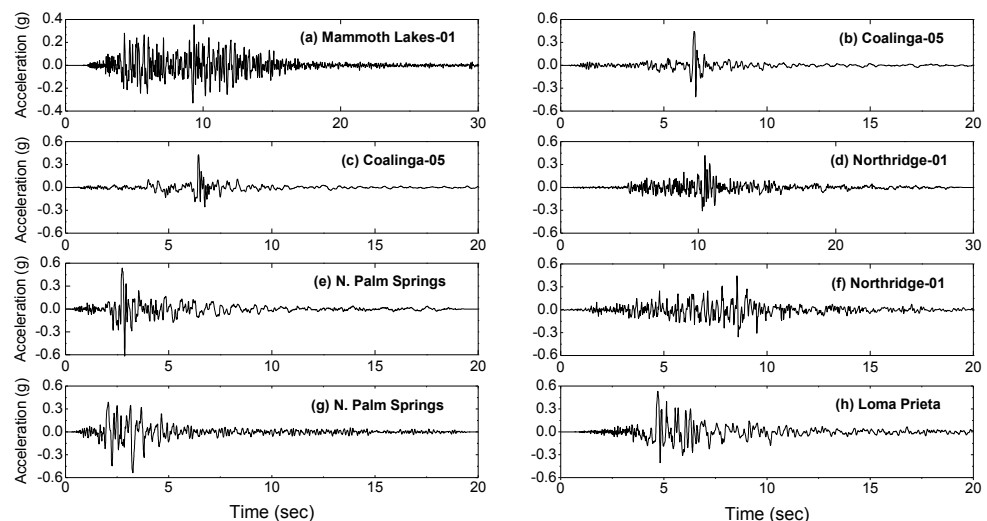

**Figure 11.** Eight time history recommendations for Taichung with DSHA calculations and the NGA strong-motion database




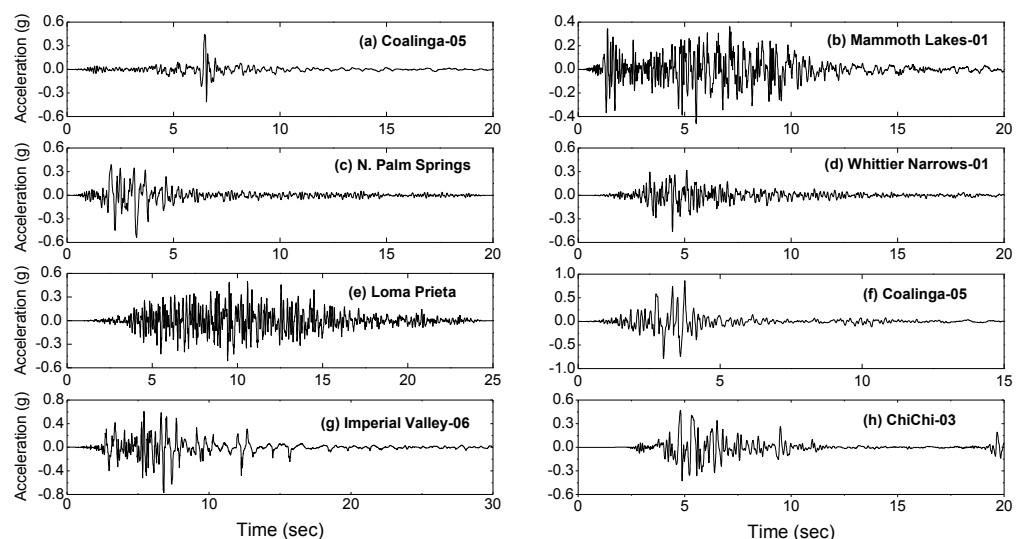

**Figure 12.** Eight time history recommendations for Chaiyi with DSHA calculations
and the NGA strong-motion database





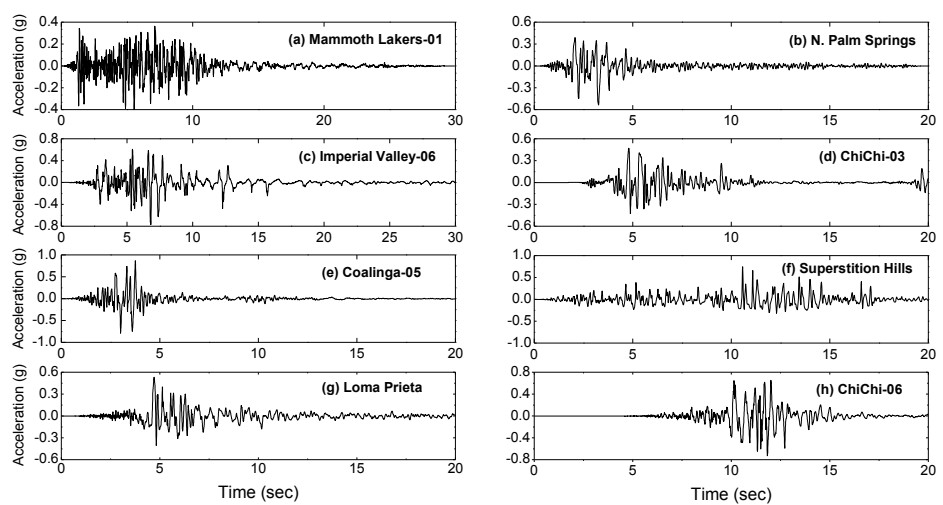

**Figure 13.** Eight time history recommendations for Hualien with DSHA calculations
and the NGA strong-motion database





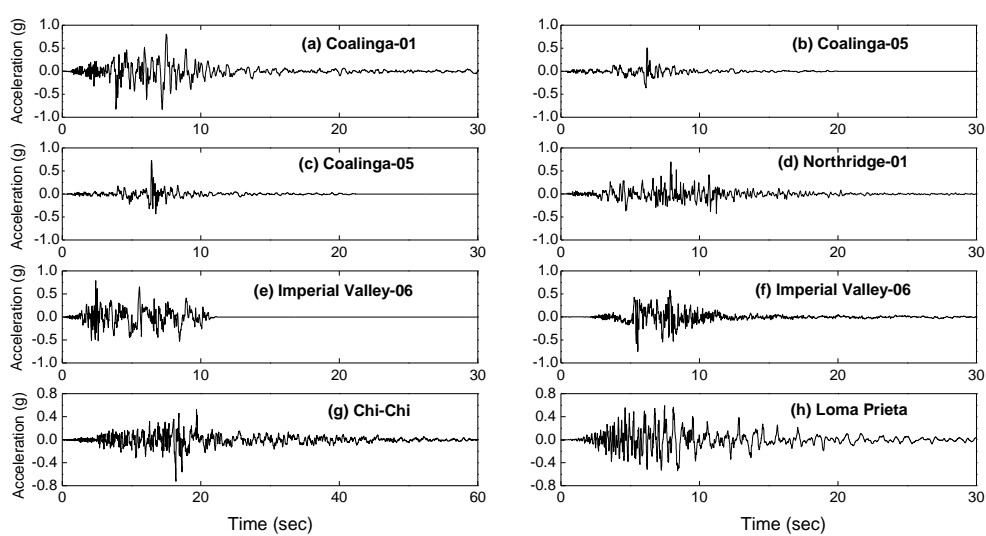

**Figure 14.** Eight time history recommendations for Pingtung with DSHA calculations
and the NGA strong-motion database





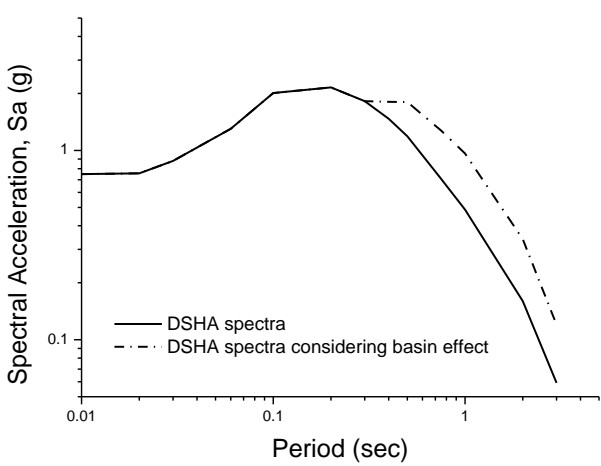

**Figure 15.** The basin effect in Taipei on response spectra; the spectra scaling follows
the suggestions of Solokov et al. (2009, 2010)

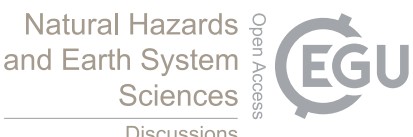

**Table 1.** Summary of Maximum Earthquake Magnitudes (in $M_w$) of Each Seismic Source around Taiwan

| Area source | Max. magnitude | Line source (active fault) | Max. magnitude | Line source (active fault) | Max. magnitude | Line source (active fault) | Max. magnitude |
|---|---|---|---|---|---|---|---|
| Zone A | 6.5 | Huangchi | 7.0 | Chenyulanchi | 7.0 | Lishan | 6.9 |
| Zone B | 6.5 | Hsiaoyukeng | 7.0 | Chiuchiungkeng | 7.0 | Meilun | 7.3 |
| Zone C | 7.1 | Sanchiao | 7.0 | Kukeng | 6.3 | Ueimei | 7.5 |
| Zone D | 7.3 | Nankan | 6.5 | Meishan | 6.5 | Yuli | 7.5 |
| Zone E | 7.3 | Shuanglienpo | 6.2 | Chukou | 7.5 | Chihshang | 7.3 |
| Zone F | 7.3 | Yangmei | 6.6 | Muchiliao | 7.1 | Yuli west | 7.3 |
| Zone G | 6.5 | Hukou | 6.9 | Liuchia | 7.1 | Luyeh | 6.9 |
| Zone H | 7.3 | Hsinchu | 6.8 | Tsochen | 6.4 | Lichi | 7.1 |
| Zone I | 6.5 | Tapingti | 6.5 | Hsinhua | 6.4 | Chimei | 7.2 |
| Zone J | 6.5 | Chutung | 6.5 | Houchiali | 6.4 | | |
| Zone K | 6.5 | Hsincheng | 6.7 | Hsiaokangshan | 6.5 | | |
| Zone L | 7.3 | Touhuanping | 6.7 | Chishan | 7.3 | | |
| Zone M | 6.5 | Seztan | 6.8 | Yuchang | 6.4 | | |
| Zone N | 8.0 | Shinchoshan | 6.5 | Yenwu | 6.7 | | |
| Zone O | 8.3 | Tuntzechiao | 6.5 | Fenshan | 6.7 | | |
| Zone P | 7.8 | Sanyi | 6.9 | Liukuei | 6.7 | | |
| Zone Q | 7.8 | Chelungpu | 7.7 | Chaochou | 7.3 | | |
| Zone R | 7.8 | Changhua | 7.6 | Hengchun | 7.2 | | |
| Zone S | 8.0 | T-H* | 7.4 | Ilan | 6.9 | | |
| Zone T | 7.8 | Shuilikeng | 7.0 | Chiaochi | 6.8 | | |

\* T-H: the Tamopu-Hsuangtung Fault





**Table 2.** Summary of the Coefficients of the Local Ground Motion Models used in This Study (Lin et al. 2011)

| Periods (sec) | $c_1$ | $c_2$ | $c_3$ | $c_4$ | $c_5$ | $\sigma_{lnY}$ |
|---|---|---|---|---|---|---|
| PGA | -3.248 | 0.943 | -1.471 | 0.1 | 0.648 | 0.628 |
| 0.01 | -3.008 | 0.905 | -1.451 | 0.11 | 0.638 | 0.623 |
| 0.06 | -1.994 | 0.809 | -1.5 | 0.251 | 0.518 | 0.686 |
| 0.09 | -1.408 | 0.765 | -1.551 | 0.28 | 0.51 | 0.709 |
| 0.1 | -1.508 | 0.785 | -1.551 | 0.28 | 0.5 | 0.713 |
| 0.2 | -3.226 | 0.87 | -1.211 | 0.045 | 0.708 | 0.687 |
| 0.3 | -4.05 | 0.999 | -1.205 | 0.03 | 0.788 | 0.657 |
| 0.4 | -5.293 | 1.165 | -1.167 | 0.011 | 0.958 | 0.655 |
| 0.5 | -6.307 | 1.291 | -1.134 | 0.0042 | 1.118 | 0.653 |
| 0.6 | -7.209 | 1.395 | -1.099 | 0.0016 | 1.258 | 0.642 |
| 0.75 | -8.309 | 1.509 | -1.044 | 0.0006 | 1.408 | 0.651 |
| 1 | -9.868 | 1.691 | -1.004 | 0.0004 | 1.485 | 0.677 |
| 2 | -12.806 | 2.005 | -0.975 | 0.0005 | 1.528 | 0.759 |
| 3 | -13.886 | 2.099 | -1.077 | 0.0004 | 1.548 | 0.787 |



**Table 3.** Summary of the Site's Coordinates, along with Respective Controlling Seismic Sources for Each Site in DSHA Computations

| City | Latitude (° N) | Longitude (° E) | Controlling source | Maximum magnitude |
|---|---|---|---|---|
| Taipei | 25.05 | 121.50 | Zone C | 7.1 |
| Kaohsiung | 22.63 | 120.32 | Zone G | 6.5 |
| Taichung | 24.15 | 120.68 | Zone E | 7.3 |
| Chiayi | 23.47 | 120.44 | Zone F | 7.3 |
| Hualien | 23.98 | 121.56 | Zone O | 8.3 |
| Pingtung | 22.02 | 120.75 | Zone L | 7.3 |


**Table 4.** Summary of the Earthquake Time History Recommendations from the NGA Database with DSHA Calculations

| City | Earthquake motion | Year | Magnitude | Rupture Distance (km) | Station | Fault Mechanism | $D_{5-95}$ (s) | $V_{s30}$ (m/s) | Scale Factor |
|---|---|---|---|---|---|---|---|---|---|
| Taipei | Mammoth Lakes-01 | 1980 | 6.06 | 4.0 | Convict Creek | N-O*** | 9.1 | 338 | 1.67 |
| | Coalinga-05 | 1983 | 5.77 | 16.1 | Pleasant Valley P.P.-FP | Reverse | 5.0 | 257 | 1.93 |
| | N. Palm Springs | 1986 | 6.06 | 6.0 | Whitewater Trout Farm | R-O** | 5.1 | 345 | 1.67 |
| | N. Palm Springs | 1986 | 6.06 | 11.2 | North Palm Springs | R-O** | 5.6 | 345 | 1.48 |
| | Coalinga-05 | 1983 | 5.77 | 16.1 | Pleasant Valley P.P.-FN | Reverse | 5.0 | 257 | 2.00 |
| | Imperial Valley-06 | 1979 | 6.53 | 2.7 | Bonds Corner | Reverse | 9.7 | 223 | 1.05 |
| | Northridge-01 | 1994 | 6.69 | 28.3 | LA – Centinela St. | Reverse | 13.0 | 235 | 0.98 |
| | N. Palm Springs | 1986 | 6.06 | 6.0 | Whitewater Trout Farm | R-O** | 5.1 | 345 | 1.52 |
| Taipei (with basin effect) | Northridge-01 | 1994 | 6.69 | 14.7 | Canoga Park | Reverse | 11.1 | 268 | 0.50 |
| | Chi-Chi | 1999 | 7.62 | 10.0 | CHY101 | R-O** | 29.0 | 259 | 1.16 |
| | Northridge-01 | 1994 | 6.69 | 28.3 | LA – Centinela St. | Reverse | 13.0 | 235 | 0.98 |
| | Coalinga-01 | 1983 | 6.36 | 8.4 | Pleasant Valley P.P. | Reverse | 8.0 | 257 | 1.32 |
| | Loma Prieta | 1989 | 6.93 | 15.2 | Capitola | R-O** | 14.7 | 289 | 1.50 |
| | Coalinga-05 | 1983 | 5.77 | 16.1 | Pleasant Valley P.P. | Reverse | 5.0 | 257 | 1.22 |
| | Imperial Valley-06 | 1979 | 6.53 | 2.7 | Bonds Corner | Strike-Slip | 9.7 | 223 | 1.35 |
| | M. - N.* -01 | 1972 | 6.24 | 4.1 | Managua- ESSO | Strike-Slip | 9.0 | 289 | 2.00 |
| Kaohsiung | Big Bear-01 | 1992 | 6.46 | 9.4 | Big Bear Lake | Strike-Slip | 10.5 | 338 | 2.32 |
| | Whittier Narrows-01 | 1987 | 5.99 | 14.5 | Garvey Res | R-O** | 5.9 | 468 | 2.49 |
| | Loma Prieta | 1989 | 6.93 | 10 | Gilroy-Gavilan Coll | R-O** | 4.7 | 729 | 2.04 |
| | Coalinga-05 | 1983 | 5.77 | 16.1 | Pleasant Valley P.P. | Reverse | 4.9 | 257 | 1.90 |
| | N. Palm Springs | 1986 | 6.06 | 6.8 | Desert Hot Springs | R-O** | 7.1 | 345 | 2.50 |
| | Loma Prieta | 1989 | 6.93 | 14.7 | Santa Teresa Hills | R-O** | 10 | 271 | 2.50 |
| | Mammoth Lakes-01 | 1980 | 6.06 | 4 | Convict Creek | N-O*** | 9.1 | 338 | 1.60 |
| | N. Palm Springs | 1986 | 6.06 | 11.2 | North Palm Springs | R-O** | 5.6 | 345 | 1.46 |



**Table 4.** Summary of the Earthquake Time History Recommendations from the NGA Database with DSHA Calculations (Continued-I)

| City | Earthquake motion | Year | Magnitude | Rupture Distance (km) | Station | Fault Mechanism | $D_{5-95}$ (s) | $V_{s30}$ (m/s) | Scale Factor |
|---|---|---|---|---|---|---|---|---|---|
| | Mammoth Lakes-01 | 1980 | 6.06 | 6.6 | Convict Creek | N-O** | 9.1 | 338 | 1.69 |
| | Coalinga-05 | 1983 | 5.77 | 16.1 | Pleasant Valley P.P.-FP | Reverse | 4.9 | 257 | 1.96 |
| | Coalinga-05 | 1983 | 5.77 | 16.1 | Pleasant Valley P.P.-FN | Reverse | 5.0 | 257 | 1.99 |
| Taichung | Northridge-01 | 1994 | 6.69 | 28.3 | LA – Centinela St. | Reverse | 11.9 | 235 | 1.99 |
| | N. Palm Springs | 1986 | 6.06 | 16.1 | North Palm Springs | R-O** | 5.6 | 345 | 1.51 |
| | Northridge-01 | 1994 | 6.69 | 22.5 | LA-UCLA | Reverse | 9.4 | 398 | 2.00 |
| | N. Palm Springs | 1986 | 6.06 | 6.0 | Whitewater Trout Farm | R-O** | 25.8 | 345 | 1.70 |
| | Loma Prieta*** | 1989 | 6.93 | 12.8 | Gilroy Array #3 | R-O** | 7.7 | 349 | 1.63 |
| | Coalinga-05 | 1983 | 5.77 | 2.7 | Pleasant Valley P.P | Reverse | 4.9 | 257 | 1.92 |
| | Mammoth Lakes-01 | 1980 | 6.06 | 6.6 | Convict Creek | N-O** | 9.1 | 338 | 1.66 |
| | N. Palm Springs | 1986 | 6.06 | 6.0 | Whitewater Trout Farm | R-O** | 5.1 | 345 | 1.67 |
| Chiayi | Whittier Narrows-01 | 1994 | 6.69 | 28.3 | LA – Obregon Park | R-O** | 7.8 | 349 | 2.00 |
| | Loma Prieta | 1989 | 6.93 | 17.5 | WAHO | R-O** | 11.1 | 376 | 1.30 |
| | Coalinga-05*** | 1983 | 5.77 | 8.5 | Oil City | Reverse | 2.8 | 376 | 1.03 |
| | Imperial Valley-06 | 1979 | 6.53 | 2.7 | Bonds Corner | Reverse | 9.7 | 223 | 1.04 |
| | Chi-Chi-03 | 1989 | 6.2 | 7.6 | TCU078 | Reverse | 6.7 | 443 | 1.66 |
| | Mammoth Lakes-01 | 1980 | 6.06 | 6.6 | Convict Creek | N-O** | 9.1 | 338 | 2.01 |
| | N. Palm Springs | 1986 | 6.06 | 6.0 | Whitewater Trout Farm | R-O** | 5.1 | 345 | 2.00 |
| | Imperial Valley-06 | 1979 | 6.53 | 2.7 | Bonds Corner | Reverse | 9.7 | 223 | 1.26 |
| Hualien | Chi-Chi-03 | 1989 | 6.2 | 7.6 | TCU078 | Reverse | 6.7 | 443 | 2.00 |
| | Coalinga-05*** | 1983 | 5.77 | 8.5 | Oil City | Reverse | 2.8 | 376 | 1.24 |
| | Superstition Hills-02 | 1987 | 6.54 | 5.6 | Superstition Camera | Strike-Slip | 12.1 | 362 | 1.53 |
| | Loma Prieta*** | 1989 | 6.93 | 12.8 | Gilroy Array #3 | R-O** | 7.7 | 349 | 1.92 |
| | Chi-Chi-06 | 1989 | 6.3 | 10.1 | TCU079 | Reverse | 4.0 | 443 | 1.28 |




**Table 4.** Summary of the Earthquake Time History Recommendations from the NGA Database with DSHA Calculations (Continued-II)

| City | Earthquake motion | Year | Magnitude | Rupture Distance (km) | Station | Fault Mechanism | $D_{5-95}$ (s) | $V_{s30}$ (m/s) | Scale Factor |
|------|-------------------|------|-----------|-----------------------|---------|-----------------|-----------|-----------|--------------|
|  | Imperial Valley-06 | 1979 | 6.53 | 0.3 | Aeropuerto Mexicali | Strike-Slip | 7.1 | 274 | 2.02 |
|  | Imperial Valley-06 | 1979 | 6.53 | 3.9 | EL Centro Array #8 | Strike-Slip | 5.8 | 206 | 1.38 |
|  | Coalinga-01 | 1983 | 6.36 | 8.4 | Pleasant Valley P.P. | Reverse | 8.0 | 257 | 1.30 |
| Pingtung | Coalinga-05 | 1983 | 5.77 | 16.1 | Pleasant Valley P.P. | Reverse | 5.0 | 257 | 1.50 |
|  | Coalinga-05 | 1983 | 5.77 | 23.5 | Bonds Corner | Reverse | 5.0 | 257 | 1.26 |
|  | Northridge-01 | 1994 | 6.69 | 15.6 | Tarzana-Cedar Hill A | Reverse | 10.3 | 257 | 2.00 |
|  | Chi-Chi | 1999 | 7.62 | 10.0 | CHY101 | R-O** | 29.0 | 258 | 1.59 |
|  | Loma Prieta | 1989 | 6.93 | 15.2 | Capitola | R-O** | 14.7 | 288 | 1.43 |

1. * M. - N. = Managua – Nicaragua
2. R-O** = Reverse – oblique
3. N-O** = Normal – oblique
4. *** refers to pulse-like record