# Peer review of "A Procedure to Select Earthquake Time Histories for Deterministic Seismic Hazard Analysis"

_Natural Hazards and Earth System Sciences, 2017_

## Referee Comment (RC1) · Anonymous Referee #1 · 27 Mar 2017

The common procedures for DSHA and GM-selection were applied in the case studies for six Taiwan sites (cities). The review comments are listed as followings: 1. Based on the DSHA results, all the controlling seismic sources of the six study sites are the area sources. However, the criteria for assigning the locations (hypocenter or the rupture plane) of the earthquake scenarios of the area sources were not provided. For example, the controlling magnitudes of the study sites b and c (Kaohsiung city and Taichung city) are Mw6.5 and Mw7.3, respectively, but, the RSs are similar to each other as shown in Figures 4. It means that a shorter source-to-site distance was assigned to the site b than that to the site c. What are the criteria for assigning the locations (hypocenter or the rupture plane) of the earthquake scenarios (the worst-case?)? It

should be noted that the area source models (Cheng et al., 2007) were developed for PSHA, and might not be adequate for the DSHA. In this paper, the upper-bound magnitude of area-source zone C, Mw7.1, was used for the DSHA scenario, however, the magnitude of Mw7.1 came from a historical event occurred in the subduction zone with a focus depth more than 70km. This paper may not assign a more likely earthquake scenario for the DSHA, even for the worst-case. Similar questions can be found on the other study sites.

2. The DSHA spectra are similar to each other for most of the study sites; however, the earthquake records from the GM-selection are quite different (I was surprised by these results). For example, the RSs of the Taichung and Chiayi cities are the same. But, the GM-selection results are different. More discussions on this or providing of other detail conditions of the GM-selection would be helpful.

3. Furthermore, it seems that the RSs (as shown in Figures 4) of the study sites were generated from the "attenuations for the hanging-wall and rock sites (Lin et al. 2011)", not the ones shown in Table 2 (for hanging-wall and soil sites). I suppose that this minor mistake is not important, but a correction of Table 2 will be better and appreciated. And, do you think the specific hanging-wall attenuations are good for the area sources? It's questionable for the cases with very short distance.

---

## Short Comment (SC1) · 20 May 2017

The discussion paper presents a general method to select ground motion time histories in Taiwan region. The findings and results presented through this study may have the potential to be of high impact for the study region. I have some comments provided as follows:

1. I recommend removing the term "site‐specific' appears many times in the manuscript (e.g. Page 8, Line 201). The term implies consideration of local site conditions. However, for all cases the authors used a single ground motion prediction relationship. The term "site-specific" in Taiwan may include quite different geological conditions, such as, alluvium filled basin (Taipei and Ilan basin); thick Quaternary

strata (Chianan Plain); relatively stiff soils in extended hilly areas (north–western part of the Island). Moreover, the upper part of the deposits may be characterized by large spatial variations of thickness and geotechnical characteristics (e.g. Taipei basin). Thus, to my mind, it would be better to construct target response spectra using semi–empirical approach, i.e. applying stochastic simulation based on spectral seismological models together with corresponding site amplifications.

2. The strong motion duration, Arias intensity and other parameters are also important and should be considered in ground motion selecting for engineering practices in Taiwan. These criteria can be found in some standards (ASCE 4–98, ASCE/SEI 43–05, NRC/RG 1.208, . . .). It is recommended to show these parameters in the selection process for the selected records.

3. lines 214-235. The "Chi-Chi Earthquake's motions are not selected" section is not relevant with this study and thus it should be removed. Besides, the "basin effect" paragraph is very confusing. In other words, it is not understandable how the basin effect in the calculation chain is introduced and how the geographical distribution of these basins is considered. Please clarify it.

---

## Author Comment (AC2) · 6 Jun 2017

The authors thank for the comment and discussion on the manuscript. Replies to the comment are provided as follows.

The discussion paper presents a general method to select ground motion time histories in Taiwan region. The findings and results presented through this study may have the potential to be of high impact for the study region. I have some comments provided as follows:

1. I recommend removing the term "site-specific' appears many times in the manuscript

[Figure]

(e.g. Page 8, Line 201). The term implies consideration of local site conditions. However, for all cases the authors used a single ground motion prediction relationship. The term "site-specific" in Taiwan may include quite different geological conditions, such as, alluvium filled basin (Taipei and Ilan basin); thick Quaternary strata (Chianan Plain); relatively stiff soils in extended hilly areas (north western part of the Island). Moreover, the upper part of the deposits may be characterized by large spatial variations of thickness and geotechnical characteristics (e.g. Taipei basin). Thus, to my mind, it would be better to construct target response spectra using semi empirical approach, i.e. applying stochastic simulation based on spectral seismological models together with corresponding site amplifications.

Reply: The authors appreciate the comments. We refer time-history recommendations in this study as "tentative site-specific" because the site effect is not comprehensively characterized with a more detailed site response analysis, but with a soil-site ground motion prediction model. Therefore, the selected ground motion time-histories could be recommended for general earthquake analytical cases, where specific site investigations are not performed. Following this line, we should modify the Subtitle 4.2 in the manuscript to "Tentative site-specific time histories". Actually we have some contents in the manuscript clarifying "site-specific" ground motions provided in Lines 202-213, Page 8:

"This paper presents an option to select earthquake time histories from the reputable NGA database. But strictly speaking, those time history recommendations are not site-specific, because the site condition is not carefully taken into account with a comprehensive site investigations and site response analyses. In other words, the site-specific motions are those from seismic hazard analyses, to site response studies (e.g., Du and Pan, 2016). As a result, this study refers to those time-history recommendations as "tentative site-specific," because the site effect is not comprehensively characterized with a more detailed site response analysis, but with a soil-site ground motion prediction model. Therefore, the selected ground motion time-histories could be recommended for general earthquake analytical cases, where specific site investigations are not performed. Since the recommended time-histories can reasonably reflect the local seismic hazards at these cities, they should be used as basic results and then be serviceable for common engineering practice."

2. The strong motion duration, Arias intensity and other parameters are also important and should be considered in ground motion selecting for engineering practices in Taiwan. These criteria can be found in some standards. It is recommended to show these parameters in the selection process for the selected records.

Reply: Comments appreciated. The authors fully agree that spectral shape only provides a partial picture of real ground-motion characteristics and doesn't contain other important features such as Arias intensity, duration as well as ground-motion nonstationarity. These parameters have been found to be important in the analysis of certain types of structures and may be incorporated in ground-motion selection process if possible. Following this line, the author recently proposes a new technique to generate synthetic ground motions that not only match the target response spectrum, but also match target Arias intensity build-up and significant duration (Huang and Wang, 2017). On the other hand, this paper aims at presenting a detailed procedure in selecting ground-motion time histories from a reputable global database for major sites in Taiwan with the use of an interactive tool when comprehensive seismic hazard assessments and site investigations are yet available. Multiple-target ground motion selection is somewhat beyond the scope of the current study, and we can add discussion regarding this concern in the revised manuscript.

3. lines 214-235. The "Chi-Chi Earthquake's motions are not selected" section is not relevant with this study and thus it should be removed. Besides, the "basin effect" paragraph is very confusing. In other words, it is not understandable how the basin effect in the calculation chain is introduced and how the geographical distribution of these basins is considered. Please clarify it.

Reply: Comments appreciated. Regarding the basin effect of Taipei city, we follow several previous studies (Sokolov et al., 2000, 2009) on the topic to scale the target response spectra at a specific range of periods, with such a site effect properly considered. Following the suggestions, we upscale spectral accelerations at long period to incorporate the basin effect. Consequently, the time histories matching the up-scaled response spectrum are selected from the NGA database.

References:

Du, W., and Pan, T. C. (2016). Site response analyses using downhole arrays at various seismic hazard levels of Singapore, Soil Dynamics and Earthquake Engineering, 90, 169-182.

Huang, D., and Wang, G. (2017). Energy-compatible and spectrum-compatible (ECSC) ground motion simulation using wavelet packets, Earthquake Engineering & Structural Dynamics, published online, DOI 10.1002/eqe.2887.

Sokolov, V., Loh, C. H., and Wen, K. L. (2000). Empirical study of sediment-filled basin response: The case of Taipei city, Earthquake Spectra, 16, 681-787.

Sokolov, V., Wen, K. L., Miksat, J., Wenzel, F., and Chen, C. T. (2009). Analysis of Taipei basin response for earthquakes of various depths, Terrestrial, Atmospheric and Oceanic Sciences, 20, 687-702.

---

## Referee Comment (RC2) · Anonymous Referee #2 · 15 Jun 2017

The manuscript is presenting a procedure for selecting the acceleration time histories from the New Generation Attenuation database for six cities in Taiwan needed for either earthquake resistance design or assessing the seismic performance of existing structures. 1- In page 3 the second step for DSHA is to define the Mmax and closest distance to each fault, how the author did assign the distance between source and sites of interest? I did not see any location for the maximum magnitude earthquakes assigned to each fault line or area. 2- There is no information about the fault style or fault type for each fault line or area which could be important in selection criteria. 3- Lines 66-67, simple method does not mean accurate estimate and with new updates of DSHA and PSHA it is easy to resolve the transparency issue. 4- Why did the author

not use PSHA or Neo-DSHA (Magrin et al. 2016 and references therein) in the estimation of the target response spectra, since they are incorporating sets of earthquakes scenarios and the resulted RSs will cover a wide range of possible scenarios than only on worst-case scenario used by this study? The long period part of the RSs might be dominated by far field sources which can be easily tackled by incorporating sets of scenarios. Keep in mind for PSHA weights in a logic tree were commonly determined by a large group of experts instead of "the author's experience and judgment". 5- Lin et al.(2011) attenuation model is not the only available local model for Taiwan, so why the authors employed only one model rather than carrying out a sensitivity analysis in order to better assess the epistemic uncertainty?. 6- Lin et al (2011) have developed different attenuation relations based on source characteristics and site effects, what is the basis for the selection of the attenuation relation being used i.e. the hanging-wall and soil sites relation, for estimating the RSs for the six cities, do all the sites are located on the hanging wall? "Chi-Chi earthquake records are used in developing the hanging-wall/footwall attenuation relation" see Lin et al (2011). Please explain this in the proper place in the manuscript. 6- The authors used Lin et al. (2011) attenuation relationship in order to predict the response spectra for periods ranging from 0.01s - 5s, but in table 2 and the response spectra figures, the author present periods only till 3 seconds. 7- How did the authors decide the scaling factor range? Please explain in the manuscript. 8- The authors described the duration of the ground motion and fault style as casual parameters whereas they are very important for the selection of proper time histories. 9- In Figure 4 the response spectra for the first 5 cities are almost the same, do the authors expect big differences in the selected time histories if yes, why?. Is it ok to use one RS for the 5 cities? 10- Lines 224-225 "First, the adopted local GMPE was developed with 42 earthquakes, 85% of which 225 are not associated with the Chi-Chi earthquake, its foreshocks and aftershocks (Lin et al., 2011)" this is not correct because in that study 44 local earthquakes and 8 earthquakes from outside Taiwan were used and about 81% of which are not associated with Chi-Chi event. I do not know if the authors recognised that about 52% of the records used to develop the attenuation

relations of Lin et al. (2011) are coming from Chi-Chi 1999 earthquake and its after-shocks. 11- In the table4 there are two Chi-Chi earthquakes (i.e. 1989 and 1999), in the text, the authors used only Chi-Chi earthquake without any additional information about the year or magnitude, it is better to be more specific. 12- Line 221 "Why Chi-Chi earthquake's motions are not selected?" this title does not consist with what is mentioned in Table 4, where Chi-Chi, 1999 earthquake has been selected for Taipei (with basin effect) and Pingtung cities. 13- Lines 229 and 230 "The second reason is that the employed searching process does not specify more weights or 230 preferences to local earthquakes" actually the selection of local earthquake records will be better since they intrinsically contain the correct path effects which can affect the experienced ground motion at the site of interest. 14- Lines 233-235 are unclear, please re-write.

References Magrin, A., Gusev, A. A., Romanelli, F., Vaccari, F., & Panza, G. F. (2016). Broadband NDSHA computations and earthquake ground motion observations for the Italian territory. International Journal of Earthquake and Impact Engineering, 1(1-2), 131-158.‏ Lin, P. S., Lee, C. T., Cheng, C. T., and Sung, C. H.: Response spectral attenuation relations for shallow crustal earthquakes in Taiwan, Eng. Geol., 121, 150-164, 2011.

---

## Author Response (AR1)

**Reply to Anonymous Referee #1**

The authors thank for the feedback. We appreciate the comments and valuable discussions on this manuscript. The comments and replies are provided as follows.

1. The common procedures for DSHA and GM-selection were applied in the case studies for six Taiwan sites (cities). The review comments are listed as followings: 1. Based on the DSHA results, all the controlling seismic sources of the six study sites are the area sources. However, the criteria for assigning the locations (hypocenter or the rupture plane) of the earthquake scenarios of the area sources were not provided. For example, the controlling magnitudes of the study sites b and c (Kaohsiung city and Taichung city) are Mw6.5 and Mw7.3, respectively, but, the RSs are similar to each other as shown in Figures 4. It means that a shorter source-to-site distance was assigned to the site b than that to the site c. What are the criteria for assigning the locations (hypocenter or the rupture plane) of the earthquake scenarios (the worst-case?)? It should be noted that the area source models (Cheng et al., 2007) were developed for PSHA, and might not be adequate for the DSHA. In this paper, the upper-bound magnitude of area-source zone C, Mw7.1, was used for the DSHA scenario, however, the magnitude of Mw7.1 came from a historical event occurred in the subduction zone with a focus depth more than 70km. This paper may not assign a more likely earthquake scenario for the DSHA, even for the worst-case. Similar questions can be found on the other study sites.

**Reply:** There was indeed a mistake about the DSHA calculation of Kaohsiung city, which has a controlling source Zone G with a maximum considered earthquake of M6.5. Therefore, the response spectra based on DSHA computation scheme for Kaohsiung city should be smaller than that of Taichung city, instead of the similar trend as pointed by the reviewer. The updated computation of DSHA-based response spectrum and recommended ground-motion waveforms for Kaohsiung city will be provided in the revised manuscript.

Also thanks for pointing out that the maximum magnitude $M_w$ 7.1 of area source C came from a historical subduction event. Nonetheless, the seismic zonation used in this study (from Zone A to Zone T) is categorized as shallow crustal regional source following previous researchers' work (i.e., Tsai 1986; Cheng et al. 2007). The maximum earthquake magnitude reflects a combined effect of regional seismology regarding historical earthquakes, focal mechanism, and source zonation, etc. Thus, the maximum magnitude of these seismogenic zones (e.g. M7.1 for source C) is adopted as the worst-case scenario during DSHA calculations. The worst-case scenario was used for identifying the earthquake scenario considered in DSHA analysis; for each area source considered, the closest source-to-site distance is assigned accordingly, as listed in the updated Table 3 below. More discussions on the worst-case scenario for each study city will also be provided in the revised manuscript.

**Table 3 (updated).** Summary of the Site's Coordinates, along with Respective Controlling Seismic Sources for Each Site in DSHA Computations

| City | Latitude (° N) | Longitude (° E) | Controlling source | Maximum magnitude | Closest source-to-site distance (km) |
|---|---|---|---|---|---|
| Taipei | 25.05 | 121.50 | Zone C | 7.1 | 2 |
| Kaohsiung | 22.63 | 120.32 | Zone G | 6.5 | 2 |
| Taichung | 24.15 | 120.68 | Zone E | 7.3 | 2 |
| Chiayi | 23.47 | 120.44 | Zone F | 7.3 | 2 |
| Hualien | 23.98 | 121.56 | Zone O | 8.3 | 2 |
| Pingtung | 22.02 | 120.75 | Zone L | 7.3 | 2 |

2. The DSHA spectra are similar to each other for most of the study sites; however, the earthquake records from the GM-selection are quite different (I was surprised by these results). For example, the RSs of the Taichung and Chiayi cities are the same. But, the GM-selection results are different. More discussions on this or providing of other detail conditions of the GM-selection would be helpful.

**Reply:** Thanks for this valuable comment. In the procedure described in this study, ground-motion time histories are selected according to a quantitative measure, the mean squared error (MSE), which evaluates how well a time history conforms to the target spectrum. The DGML search engine used in this study searches the NGA database for ground-motion waveforms that satisfy the general criteria (i.e. $5.5 < M_w < 8$, $0 < R_{rup} < 30$ km) and then ranks theses records in an order of increasing MSE. It means that the ground-motion waveform that matches the target RS best has the lowest MSE and will be ranked No. 1. To be more specific, the MSE is defined using the following equation (Wang *et al.* 2015):

$$\mathrm{MSE} = \frac{\sum_i w(T_i) \left\{ \ln\left(Sa^{t\arg et}(T_i)\right) - \ln\left(f \times Sa^{record}(T_i)\right) \right\}^2}{\sum_i w(T_i)} \tag{1}$$

where $T_i$ denotes considered spectral periods, $w(T_i)$ denotes a weight function that allows for assigning weights to different period ranges so that the periods of more interest can be emphasized in the ground-motion selection process, $f$ represents a scale factor to linearly scale the whole ground-motion time history. More detailed condition on how ground motions are selected will be added in the revised manuscript. It should be also noted that the MSE does not vary too much in some cases. For example, as highlighted in the following Figure 1, the MSE ranges from 0.023-0.035, indicating that the selected scaled ground motions are almost equally good and compatible with the target response spectrum. Therefore, in this study, we intentionally select some other ground-motion waveforms if some of them have been recommended in the other study cities. As a result, different GM selection results are recommended for the Taichung and Chiayi cities although they have the similar target response spectra. We expect, by doing so,

more flexibility and options could be provided for time-history analyses in engineering practice. It should be also mentioned that although different ground motions are selected for various sites, they are statically consistent and compatible with the corresponding DSHA spectrum.

[Figure]

**Figure 1.** The screenshot of the database's interface. The red box highlights the column that reports the computed MSEs of selected ground-motion records.

3. Furthermore, it seems that the RSs (as shown in Figures 4) of the study sites were generated from the "attenuations for the hanging-wall and rock sites (Lin et al. 2011)", not the ones shown in Table 2 (for hanging-wall and soil sites). I suppose that this minor mistake is not important, but a correction of Table 2 will be better and appreciated. And, do you think the specific hanging-wall attenuations are good for the area sources? It's questionable for the cases with very short distance.

**Reply:** Thanks for pointing out the typo and meaningful discussion. The attenuation adopted in this study is indeed for the hanging-wall and rock sites, and thus Table 2 is updated as follows. For the second concern, we agree that the worst-case scenarios considered in this manuscript may not be the hanging wall case. However, since the Lin et al. (2011) model is the only available regional-specific response spectral attenuation model for shallow crustal earthquakes to the authors' best knowledge, this hanging-wall attenuation model is then adopted in the current study with reasonably conservative results provided. Besides, to avoid possible saturation at

short distance in the attenuation model, each seismogenic area source was defined with assumed depth as 2 km.

**Table 2 (updated).** Summary of the Coefficients of the Local Ground Motion Models used in This Study (Lin et al. 2011)

| Periods (s) | $c_1$ | $c_2$ | $c_3$ | $c_4$ | $c_5$ | $\sigma_{\ln Y}$ |
|---|---|---|---|---|---|---|
| PGA | -3.279 | 1.035 | -1.651 | 0.152 | 0.623 | 0.651 |
| 0.01 | -3.253 | 1.018 | -1.629 | 0.159 | 0.612 | 0.647 |
| 0.06 | -1.738 | 0.908 | -1.769 | 0.327 | 0.502 | 0.702 |
| 0.09 | -1.237 | 0.841 | -1.750 | 0.478 | 0.402 | 0.748 |
| 0.1 | -1.103 | 0.841 | -1.765 | 0.455 | 0.417 | 0.750 |
| 0.2 | -2.767 | 0.980 | -1.522 | 0.097 | 0.627 | 0.697 |
| 0.3 | -4.440 | 1.186 | -1.438 | 0.027 | 0.823 | 0.685 |
| 0.4 | -5.630 | 1.335 | -1.414 | 0.014 | 0.932 | 0.683 |
| 0.5 | -6.746 | 1.456 | -1.365 | 0.006 | 1.057 | 0.678 |
| 0.6 | -7.637 | 1.557 | -1.348 | 0.0033 | 1.147 | 0.666 |
| 0.75 | -8.641 | 1.653 | -1.313 | 0.0015 | 1.257 | 0.652 |
| 1 | -9.978 | 1.800 | -1.286 | 0.0008 | 1.377 | 0.671 |
| 2 | -12.611 | 2.058 | -1.261 | 0.0005 | 1.497 | 0.706 |
| 3 | -13.303 | 2.036 | -1.234 | 0.0013 | 1.302 | 0.702 |

**References:**

Cheng, C. T., Chiou, S. J., Lee, C. T., and Tsai, Y. B.: Study on probabilistic seismic hazard maps of Taiwan after Chi-Chi earthquake, J. GeoEngineering, 2, 19-28, 2007.

Lin, P. S., Lee, C. T., Cheng, C. T., and Sung, C. H.: Response spectral attenuation relations for shallow crustal earthquakes in Taiwan, Eng. Geol., 121, 150-164, 2011.

Tsai, Y. B.: Seismotectonics of Taiwan, Tectonophysics, 125, 17-37, 1986.

Wang, G., Youngs, R., Power, M., and Li, Z.: Design ground motion library: an interactive tool for selecting earthquake ground motions, Earthq. Spectra, 31, 617-635, 2015.

**Reply to Anonymous Referee #2**

The manuscript is presenting a procedure for selecting the acceleration time histories from the New Generation Attenuation database for six cities in Taiwan needed for either earthquake resistance design or assessing the seismic performance of existing structures.

1- In page 3 the second step for DSHA is to define the Mmax and closest distance to each fault, how the author did assign the distance between source and sites of interest? I did not see any location for the maximum magnitude earthquakes assigned to each fault line or area.

Reply: The authors thank for the comments. In this study, the seismic zonation for DSHA (from Zone A to Zone T) is categorized as shallow crustal regional source following several previous work carried out by local researchers (i.e., Tsai 1986; Cheng et al. 2007). The maximum earthquake magnitude represent a combined effect of regional seismology regarding historical earthquakes, focal mechanism, and source zonation, etc. Thus, Mmax and closest distance of these seismogenic zones (e.g. M7.1 for source C) are adopted as the worst-case scenario throughout DSHA calculations. For each area source considered, the closest source-to-site distance is added in Table 3 in the revised manuscript, as follows:

**Table 3 (revised).** Summary of the Site's Coordinates, along with Respective Controlling Seismic Sources for Each Site in DSHA Computations

| City | Latitude ($^o$ N) | Longitude ($^o$ E) | Controlling source | Maximum magnitude | Closest source-to-site distance (km) |
|---|---|---|---|---|---|
| Taipei | 25.05 | 121.50 | Zone C | 7.1 | 2 |
| Kaohsiung | 22.63 | 120.32 | Zone G | 6.5 | 2 |
| Taichung | 24.15 | 120.68 | Zone E | 7.3 | 2 |
| Chiayi | 23.47 | 120.44 | Zone F | 7.3 | 2 |
| Hualien | 23.98 | 121.56 | Zone O | 8.3 | 2 |
| Pingtung | 22.02 | 120.75 | Zone L | 7.3 | 2 |

2- There is no information about the fault style or fault type for each fault line or area which could be important in selection criteria.

Reply: Comments appreciated. We added fault type for each line source in Table 1 in the revised manuscript. The area sources are shallow crustal regional sources following previous researchers' work (i.e., Tsai 1986; Cheng *et al.* 2007).

3- Lines 66-67, simple method does not mean accurate estimate and with new updates of DSHA and PSHA it is easy to resolve the transparency issue.

Reply: Comments appreciated. We fully agree that new updates and generations of DSHA and PSHA allow for transparency for both methods. Therefore, we rewrite Lines 66-67 as follows: "Compared to the complicated probabilistic approach, DSHA is an analysis accounting for a

worst-case scenario in terms of earthquake size and location. Specifically, DSHA utilizes the maximum magnitude and shortest source-to-site distance to evaluate the ground motion intensities under such a worse-case scenario. The basic steps are listed as follows … "

4- Why did the author not use PSHA or Neo-DSHA (Magrin et al. 2016 and references therein) in the estimation of the target response spectra, since they are incorporating sets of earthquakes scenarios and the resulted RSs will cover a wide range of possible scenarios than only on worst-case scenario used by this study? The long period part of the RSs might be dominated by far field sources which can be easily tackled by incorporating sets of scenarios. Keep in mind for PSHA weights in a logic tree were commonly determined by a large group of experts instead of "the author's experience and judgment".

Reply: Thanks for the suggestions. Actually there are quite a few PSHA studies for Taiwan, including several ones the authors previously conducted (i.e. Wang, J.P. Huang, D., et al. 2013; Cheng *et al.* 2007). In this study, we intended to perform a deterministic study which allows for full consideration of the worst-case scenario for several study sites in Taiwan. We also thank the reviewer for pointing out the concern about long period part of the response spectra, which may be indeed dominated by a far field seismic source, i.e. a subduction source around Taiwan. However, in the current ground-motion selection practice, ground motions selected from the most comprehensive Next Generation Attenuation (NGA) database managed by the Pacific Earthquake Engineering Center (PEER) are all recorded from historic shallow crustal earthquakes, which have substantially different characteristics with those recorded from a subduction earthquake. To the authors' best knowledge, development of the subduction earthquake database by PEER is still ongoing and the database is yet released. In this regard, we focus on the short to moderate period of response spectra in the current study to avoid unnecessary confusion induced. `

5- Lin et al.(2011) attenuation model is not the only available local model for Taiwan, so why the authors employed only one model rather than carrying out a sensitivity analysis in order to better assess the epistemic uncertainty?.

Reply: Comments appreciated. We know that there are indeed some other local GMPEs such as Lin and Lee 2008. However, considering the above mentioned one is particularly for subduction earthquakes, we do not incorporate it in the ground-motion selection process since ground motions in the NGA database are all from shallow crustal earthquakes. Also, the worst-case scenario is used throughout the current for identifying the earthquake scenario in DSHA analysis. Thus, we do not induce the epistemic uncertainty in this concern.

6- Lin et al (2011) have developed different attenuation relations based on source characteristics and site effects, what is the basis for the selection of the attenuation relation being used i.e. the hanging-wall and soil sites relation, for estimating the RSs for the six cities, do all the sites are located on the hanging wall? "Chi-Chi earthquake records are used in developing the hanging-wall/footwall attenuation relation" see Lin et al (2011). Please explain this in the proper place in the manuscript.

Reply: Thanks for the meaningful discussion. The attenuation adopted in this study is indeed for the hanging-wall and rock sites (ref. Table 2 in the revised manuscript). We agree that some of the study sites may not be located on the hanging wall. However, the hanging-wall attenuation is adopted in the current study for the consideration of worst-case scenario for DSHA and reasonably conservative results.

6- The authors used Lin et al. (2011) attenuation relationship in order to predict the response spectra for periods ranging from 0.01s - 5s, but in table 2 and the response spectra figures, the author present periods only till 3 seconds.

Reply: Comments appreciated. In this study, we focus on periods ranging from 0.01 s to 3 s for seismic design of low to median rise buildings, with an emphasis on buildings lower than 30 stories. We do not really fit the target response spectrum over a broad range (e.g. from 0.01 s – 5 s) and expect by doing so, the fitting error may be significantly increased. It is also worth mentioning that in contemporary practice of ground-motion selection, sets of time histories are also commonly selected based on a target conditional mean spectra, which provides realistic spectral shapes for scenario earthquakes. Researchers identified that fitting the entire uniform hazard spectrum (UHS) over a wide range may be overly broad and thus overly conservative for a single earthquake, because the UHS represent a combination of multiple scenario earthquakes.

7- How did the authors decide the scaling factor range? Please explain in the manuscript.

Reply: Ground motion scaling is actually a topic subjected to intense debate over the past decades, since researchers recognized that improper scaling of a record can lead to bias estimates of structural responses (Luco and Bazzurro 2007). However, it was also reported that if the record is scaled multiple well-established target parameters (e.g. Arias intensity, PGV), ground motions can be scaled by a reasonable factor and still can result in unbiased estimates of responses for structural/geotechnical systems. Therefore, in this study, we follow the general practice of the developer of the Design Ground Motion Library (DGML), with the range of the scaling factor specified from 0.4 to 2.5 (Wang et al. 2015).

More discussion on this point is added in the revised manuscript in Lines 153-160: "….but has been subjected to intense debate over the past decades. Previous researchers pointed out that improper scaling of a record can lead to bias estimates of structural responses (Luco and Bazzurro 2007). For example, if an excessive range of scale factors is applied, the selected ground motion suite might result in drastically biased distribution of the other ground-motion characteristics, such as duration, Arias intensity, that cannot be represented by the target response spectrum. Therefore, we follow the general practice of the Design Ground Motion Library (DGML) and assign a relative narrow range of scale factors (0.4-2.5) throughout the selection procedure in this study (Wang et al. 2015)."

8- The authors described the duration of the ground motion and fault style as casual parameters whereas they are very important for the selection of proper time histories.

Reply: The authors fully agree that ground-motion duration and fault style are quite important in the selection process, so described in the original manuscript "Other causal parameters, such as

the category of fault types or the range of duration parameters, are not particularly specified". The authors used "causal" that means indicating a cause, instead of "casual" that means trivial/unconcerned parameters.

9- In Figure 4 the response spectra for the first 5 cities are almost the same, do the authors expect big differences in the selected time histories if yes, why?. Is it ok to use one RS for the 5 cities?

Reply: Thanks for the comments. The principle of deterministic seismic hazard assessment (DSHA) is to incorporate the worst-case earthquake scenario, that is, the maximum considered earthquake (MCE) occurred at the closest source-to-site distance. This so-called worst-case scenario reflects a combined effect of regional seismology regarding historical earthquakes, focal mechanism, and source zonation, etc. Thus, the maximum magnitude of these seismogenic zones (e.g. M7.1 for source C) is adopted as the worst-case scenario in the calculation. As can be seen from Table 3 in the revised manuscript, several studies sites are governed by similarly large MCE, which can result in similar spectra shape for different study sites.

10- Lines 224-225 "First, the adopted local GMPE was developed with 42 earthquakes, 85% of which are not associated with the Chi-Chi earthquake, its foreshocks and aftershocks (Lin et al., 2011)" this is not correct because in that study 44 local earthquakes and 8 earthquakes from outside Taiwan were used and about 81% of which are not associated with Chi-Chi event. I do not know if the authors recognised that about 52% of the records used to develop the attenuation relations of Lin et al. (2011) are coming from Chi-Chi 1999 earthquake and its aftershocks.

Reply: Thanks for the valuable comment. The statement about the usage of local ground-motion record in the initial submission was indeed a mistake. We removed such an argument and rewrite it in the revised manuscript in Lines 244-248:" It somewhat comes to as a surprise that the motions of the local earthquake were "out-performed" by non-local motions in matching the response spectra with local ground motion models. This might be due to two reasons. First, apart from the Chi-Chi earthquake, most events used for developing the local GMPE are not included in the NGA database. The second reason is that the employed searching process does not specify more weights or preferences to local earthquakes…."

11- In the table4 there are two Chi-Chi earthquakes (i.e. 1989 and 1999), in the text, the authors used only Chi-Chi earthquake without any additional information about the year or magnitude, it is better to be more specific.

Reply: Comment appreciated. We actually specify year, magnitude and fault rupture mechanism of each event in Table 4 in the original submission.

12- Line 221 "Why Chi-Chi earthquake's motions are not selected?" this title does not consist with what is mentioned in Table 4, where Chi-Chi, 1999 earthquake has been selected for Taipei (with basin effect) and Pingtung cities.

Reply: Comment appreciated. We modified the subtitle 4.4 in the revised manuscript to "Why local earthquake's motions are not selected for all cases?"

13- Lines 229 and 230 "The second reason is that the employed searching process does not specify more weights or preferences to local earthquakes" actually the selection of local earthquake records will be better since they intrinsically contain the correct path effects which can affect the experienced ground motion at the site of interest.

Reply: The authors agree that local ground motions may contain intrinsic correct path effects at the site of interest. However, the principle of current ground-motion selection practice is searching for time history record sets in the database on the basis of the similarity of a record's response spectral shape to a design response spectrum over a user-defined period range. Ground-motion time histories are ranked according to a quantitative measure, the mean squared error (MSE), which evaluates how well a time history conforms to the target spectrum. In such a case, local earthquake records are not always selected and recommended for the study sites because they might not perfectly conform to the target spectra.

14- Lines 233-235 are unclear, please re-write.

Reply: Comments appreciated. We rewrite the statement in the revised manuscript in Lines 248-253:" As discussed previously, the search criterion are only associated with the spectral shape, as well as seismological parameters such as magnitude, distance, site condition, etc. The search engine searches the database and ranks the records based on a quantitative measure: the mean squared error. With this in mind, as long as the size of the database is sufficient, it is not surprising that a non-local ground motion can be found better matching the target spectra."

References

Cheng, C. T., Chiou, S. J., Lee, C. T., and Tsai, Y. B.: Study on probabilistic seismic hazard maps of Taiwan after Chi-Chi earthquake, J. GeoEngineering, 2, 19-28, 2007.

Lin, P. S., Lee, C. T., Cheng, C. T., and Sung, C. H.: Response spectral attenuation relations for shallow crustal earthquakes in Taiwan, Eng. Geol., 121, 150-164, 2011.

Lin, P. S. and C. T. Lee, 2008: Ground-motion attenuation relationships for subduction-zone earthquakes in northeastern Taiwan. Bull. Seismol. Soc. Am., 98, 220-240,

Luco, N., and Cornell, C. A., 2007. Structure-specific scalar intensity measures for near-source and ordinary earthquake ground motions, Earthquake Spectra 23, 357–392.

Magrin, A., Gusev, A. A., Romanelli, F., Vaccari, F., & Panza, G. F. (2016). Broadband NDSHA computations and earthquake ground motion observations for the Italian territory. International Journal of Earthquake and Impact Engineering, 1(1-2), 131-158

Tsai, Y. B.: Seismotectonics of Taiwan, Tectonophysics, 125, 17-37, 1986.

Wang, G., Youngs, R., Power, M., and Li, Z.: Design ground motion library: an interactive tool for selecting earthquake ground motions, Earthq. Spectra, 31, 617-635, 2015.

Wang, J.P. Huang, D., Cheng, C.T., Shao, K.S., Wu, Y.C., Chang, C.W. (2013). Seismic hazard analysis for Taipei City including deaggregation, design spectra, and time history with Excel applications. Computers and Geosciences; 52, 146-154

---

## Author Response (AR2)

**Reply to the Editor**

**By Duruo Huang, Wenqi Du and Hong Zhu**

Comments to the Author:

Dear Authors,

Regarding your revisions, I feel you have mostly well replied on the reviewer´s concerns, but many of the additional information given in the reply letter is not provided in the revised manuscript. Please reorganize your paper incorporating all of your reply at appropriate positions in the text.

Referee #1 pointed out the high similarity between RS of site b and c, and in your reply you´ve indicated a mistake in the DSHA calculations for site b that should have been fixed in the revised manuscript. However, when comparing Figures 4 of the revised and original submissions I can see no difference in the RS of the sites. Please clarify this, together with a more in-depth explanation on the criteria used to assign specific earthquake locations, the general similarity of the RS, and more discussion why a source model developed for PSHA is suitable for DSHA. It should be explained why specific RS for sites a-e are required and not only one because of their similarity. Additionally, more explanation on the specific seismotectonic site characteristics of the six locations is required in the manuscript. Moreover, it is important to explain properly in the manuscript why local earthquake records were not exclusively used for this study. Adding a paragraph on this in the discussion is not sufficient, a justification why global earthquake records have been searched and why these are applicable to the Taiwan case (and site characteristics) using RS based on a local GMM should be provided in Section 3 of the manuscript.

The authors thank for the valuable comments and discussions on this manuscript. The paper has been reorganized to incorporate responses to the editor and reviewers, with changes highlighted as red fonts. The replies are provided point-by-point as follows.

**Why a source model developed for PSHA is suitable for DSHA?**

DSHA and PSHA, are commonly used methods for evaluating seismic hazard. DSHA adopts "deterministic" information during analysis, while PSHA accounts for the "probabilistic" characteristics of earthquake size, location, and ground-motion models. The seismic zonation used in this study follows previous researchers' work (i.e., Tsai 1986; Cheng 2002; Cheng et al. 2007). For example, Tsai (1986) reviewed historic seismicity data and proposed the zonation which accounts for complex local seismotectonics. The source model was developed based on historic data, site investigation and local tectonic setting. Although it has been used in the PSHA framework, it is not limited to probabilistic analysis, given that the MCE for multiple sources have been clearly specified.

Besides, tectonic setting of the six study sites are provided in Page 5, Lines 125-134: "As is well know that Taiwan is located at the boundary between the Philippine Sea Plate to the East and the Eurasian Plate to the West, the six study sites are intentionally selected to represent different geological units of the island: site (a) Taipei, site (c) Taichung and site (d) Chiayi are located at the Western Foothills, where syn-orogenic sediments of the foreland basin have been accreted and deformed.; site (b) Kaohsiung is located at the Coastal Plain as a part of the foreland basin of Taiwan; site (e) Pingtung is located within the West Central Range (or Backbone Range) with mostly Miocene to Eocene slates, corresponding to the area of highest altitudes in Taiwan; site (f) Hualien is located at the Longitudinal Valley which is believe as the suture zone between the Luzon arc and the Chinese continental margin."

*Why local earthquake records have not been exclusively used?*

The mean square error (MSE) for each single selected record has been added in Table 4 in the revised manuscript. It can be seen that the MSEs range from 0.023-0.046 for different study sites. The ground-motion waveforms have been recommended in this study based on their compatibility with the target response spectrum, and such compatibility is parametrized as the MSE.

Regarding selection of local earthquake records, we need to also look at Figure R.1, which compares spectral accelerations predicted using the local GMPE (Lin et al. 2011) with those computed using other four widely used NGA global GMPEs, namely, AS08, BA08, CB08 and CY08, for several earthquake scenarios (Abrahamson and Silva 2008; Boore and Atkinson 2008; Campbell and Bozorgnia 2008; Chiou and Youngs 2008). It can be seen that under the scenario M=7, Rrup=30 km, $Vs_{30} = 760$m/s, the spectral accelerations predicted by local attenuation agree well with the BA08 and CY08 models across a wide range of periods (i.e. from 0.01 s to 5 s). As for the scenarios M=7, Rrup=10 km, $Vs_{30} = 760$m/s, the spectral accelerations predicted by local GMPE again corresponds well with those computed using the CY08 model, as demonstrated in Figure R.1 (a). Apart from the consistency with global GMPE, it is also worth mentioning that the functional form of the local model is based on Campbell (1981), which is a quite generic and widely adopted one. Therefore, the target RS in this study is not exclusive but generalized in a way. Ground motions selected from the comprehensive NGA database based on compatibility with the target RS can be either local records or global ones, but not necessarily local motions, given the "generic target GMPE". A justification on why local earthquake records have not been exclusively used for this study is provided in the revised manuscript Page 10-11, Line 271-303.

[Figure]

Figure R.1. Comparison of spectral acceleration predicted using the local model and NGA global ground motion prediction equations (GMPE) under earthquake scenarios (a) M=7, Rrup=10 km, $Vs_{30}$ = 760m/s, (b) M=7, Rrup=30 km, $Vs_{30}$ = 760m/s; (c) M=7, Rrup=50 km, $Vs_{30}$ = 760m/s

*The criteria used to assign specific earthquake locations?*

Reply: The principle of deterministic seismic hazard assessment is to incorporate the worst-case earthquake scenario, that is, the maximum considered earthquake (MCE) occurred at the closest source-to-site distance. This worst-case scenario reflects a combined effect of regional seismology regarding historical earthquakes, focal mechanism, and source zonation, etc. Thus, the maximum magnitude of these seismogenic zones (e.g. M7.1 for source C) is adopted as the worst-case scenario in the calculation. As can be seen from Table 3 in the revised manuscript, several studies sites are governed by similarly large MCE, which can result in similar spectra shape for different study sites.

The respective controlling seismic sources for the six sites are listed in Table 3. For each site, we assumed the earthquake rupture occurred at the same site location, with a depth of 2 km. Therefore, the closest source-to-site distances are 2 km accordingly. For the earthquake scenarios considered for these sites, the only differences are the maximum magnitude assigned. For these near-fault scenarios (i.e., different Mw but the same Rrup), the resulting RSs obtained by the same local GMPE are therefore similar.

*General similarity of the RS and why specific RS for sites a-e are required and not only one because of their similarity*

Thanks for pointing out the error. The RSs for site b (Kaohsiung) in Figures 4 and 7 have been revised in the updated submission.

In this study, the response spectra for sites a-e have been developed following the framework of DSHA, which adopts the worst-case scenario accounting for historical data and focal mechanism in a region. Since several sites are governed by similar MCE, the computed response spectra are similar in such a case. However, the RS cannot be represented by a single one because of their similarity. For the site such as Hualien governed by relatively large MCE, its RS is significantly (around 30%) larger than those for other sites. On the other hand, for sites Taipei and Taichung that has quite similar MCEs as 7.1 and 7.3, respectively, Table R.1 summarizes spectral accelerations for the two sites along with their relative difference. Generally, the difference in spectral acceleration is larger than 5%, and can approach 20% at long period (i.e. 3s). Thus, specific RS for each city is required instead of using a single one to represent all cases.

Table R.1. Comparison of spectral accelerations at study sites Taipei and Taichung

| Spectral period (s) | Sa in g (Taipei) | Sa in g (Taichung) | Relative difference |
|---|---|---|---|
| 0.06 | 1.10 | 1.13 | 2.7% |
| 0.09 | 1.92 | 2.02 | 5.2% |
| 0.10 | 1.95 | 2.05 | 5.1% |
| 0.30 | 1.63 | 1.70 | 4.3% |
| 0.50 | 1.10 | 1.16 | 5.5% |
| 1.0 | 0.46 | 0.49 | 6.5% |
| 2.0 | 0.14 | 0.16 | 14.2% |
| 3.0 | 0.10 | 0.12 | 20% |

In addition to the above general observations on your revisions in the light of the referee comments, I have some additional specific comments on the presentation listed below. Please revise your paper following these and the general comments above and resubmit your paper together with a detailed point-per-point reply and a manuscript version highlighting the applied changes. After resubmission, your paper will be reviewed again by the editor.

P5L110: Here you should also refer to Figure 2 where the sites are plotted as stars (although not indicated in the Figure caption). Please also denote site letters a-f for easy identification throughout the manuscript.

Reply: Comments appreciated. We refer to "Figure 2" in Page 5 Line 124 in the revised manuscript and highlight the six study sites in the caption of Figure 2. Also, we refer study sites from (a) to (f) throughout the revised manuscript.

 If for all sites the area source is the controlling source, why do you provide the line sources and their parameters? Please explain.

Reply: We first considered and listed all possible seismic sources (i.e., area and line sources) in this region for these city sites. The worst-case scenario was then identified for each city (area source is the controlling one).

P5L120: Isn´t the source-to-site distance zero in all cases since the sites are located in the specific controlling source areas? Please explain.

Reply: Not exactly. For an area source, the upper and lower seismogenic depths are important source parameters. As the upper depth of these source areas are assumed as 2 km, the source-to-site distance is then set as 2 km accordingly.

P7L181: In Figure 6, it cannot be observed that something is highlighted. Do you refer to the red rectangle in the Figure? This is not indicated in the caption.

Reply: Comments appreciated. "MSE" (mean squared error) is indicated it in the caption of revised Figure 6. And we also added detailed description of the DGML searching interface (ref. to Figure 6) and specified functionality of the major components in Page 6 Lines 166-170 in the revised manuscript.

P8L200-P9L222: Section 4.1: I am not sure whether a general discussion on pros and cons of PSHA vs. DSHA is really required here. It would be better to argue why DSHA have been chosen for this study and how it can be developed further in the Taiwan case.

Reply: Thanks for the comment. Because in the manuscript several pervious PSHA studies for Taiwan have been referred to, it tends to be comprehensive to give a brief comparison between the two widely adopted methods. Regarding using DSHA in this study, we have following discussion in the revised manuscript Lines 241-247: "… it should come to a logical understanding that both the deterministic and probabilistic analyses are needed and useful in engineering applications. The use of the DSHA approach in this study is primarily due to its analytical simplicity and transparency. Since it has been reported that DSHA rather than PSHA is more appropriate for design of critical structures (Bommer et al., 2000), the selected ground motion suites, with a representative seismic hazard analysis and a reputable earthquake database, are then recommended for such applications."

Figure 1: This map has no scale. Reference to plotted stars (sites) is missing in the caption. Maybe you can plot more information from Table 1 in this Figure (e.g., source areas colored with max. magnitude classes).

Reply: Comments appreciated. Reference to the plotted star is added in the caption. Since Figure 1 is a schematic diagram to illustrate general computation framework of DSHA, we do not intended to refer a specific case, such as the information listed in Table 1.

Figure 2: Map has no scale. Is it needed since your DSHA only requires area sources? Please check.

Reply: Comments appreciated. We added scales in the revised Figure 2 and Figure 3. The DSHA requires both area and line sources in the computation framework. The controlling source for each site (i.e. area source in this study) was identified after the computation flow, instead of being recognized before the assessment.

Figure 6: The screenshot of the search engine is of very bad quality. Specifically, the search criteria are not visible, and the red box (probably highlighting MSE-ranges) is not indicated in the caption.

Reply: Thanks for the comments. We actually specified selection criteria in terms of magnitude bound, distance bound, Vs30, range of scale factor and weight factor, etc, in Page 7, Lines 171-188 in the revised manuscript. In addition, we added detailed description of the DGML searching interface (ref. to Figure 6) and specified functionality of the major components (Lines 166-170): "The DGML Search Engine window is shown in Figure 6. It contains the following major parts: (1) Inputs for searching criteria; (2) Prescribed range of scale factor; (3) Prescribed weight factor for spectral period; (4) Spectrum plot of selected motions; (5) MSE of each individual selected ground-motion record; (6) Scale factor of each record; (7) Event name and (8) station name of each record."

Table 1: Is it necessary to provide the parameters of the line sources?

Reply: Yes, I think it is necessary.

**References**:

Abrahamson, N. A., and Silva, W. J.: Summary of the Abrahamson & Silva NGA ground motion relations, Earthq. Spectra, 24, 67–97, 2008.

Boore, D. M., and Atkinson, G. M.: Ground-motion prediction equations for the average horizontal component of PGA, PGV, and 5% damped PSA at spectral periods between 0.01s and 10.0s, Earthq. Spectra, 24. 99–138, 2008.

Campbell, K. W., and Bozorgnia, Y.: NGA ground motion model for the geometric mean horizontal component of PGA, PGV, PGD, and 5% damped linear elastic response spectra for periods ranging from 0.01s to 10.0s, Earthq. Spectra., 24, 139–171, 2008.

Cheng, C. T.,Uncertainty analysis and de-aggregation of seismic hazard in Taiwan," Ph.D. Dissertation, Institute of Geophysics, National Central University, Chung-Li, Taiwan, 2002.

Cheng, C. T., Chiou, S. J., Lee, C. T., and Tsai, Y. B.: Study on probabilistic seismic hazard maps of Taiwan after Chi-Chi earthquake, J. GeoEngineering, 2, 19-28, 2007.

Chiou, B.S., Youngs, R.R., 2008. An NGA model for the average horizontal component of peak ground motion and response spectra. Earthquake Spectra 24, 173–215.

Lin, P. S., Lee, C. T., Cheng, C. T., and Sung, C. H.: Response spectral attenuation relations for shallow crustal earthquakes in Taiwan, Eng. Geol., 121, 150-164, 2011.

Tsai, Y.B. Seismotectonics of Taiwan, Tectonophysics, 125, 17-37, 1986.